# THE FIRST IMPRESSION PROBLEM: INTERNAL BIAS TRIGGERS OVERTHINKING IN REASONING MODELS

**Renfei Dang, Zhening Li, Shujian Huang[†], Jiajun Chen**
National Key Laboratory for Novel Software Technology, Nanjing University
{dangrf,lizn}@smail.nju.edu.cn, {huangsj,chenjj}@nju.edu.cn

## ABSTRACT

Reasoning models often exhibit overthinking, characterized by redundant reasoning steps. We identify *internal bias* elicited by the input question as a key trigger of such behavior. Upon encountering a problem, the model immediately forms a preliminary guess about the answer, which we term an internal bias since it may not be explicitly generated, and it arises without systematic reasoning. When this guess conflicts with its subsequent reasoning, the model tends to engage in excessive reflection, resulting in wasted computation. We validate the association between internal bias and overthinking across multiple models and diverse reasoning tasks. To demonstrate the causal relationship more rigorously, we conduct two counterfactual interventions, showing that removing the input question after the model reduces the redundant reasoning across various complex reasoning tasks, and manually injecting bias affects overthinking accordingly. Further interpretability experiments suggest that excessive attention to the input question serves as a key mechanism through which internal bias influences subsequent reasoning trajectories. Finally, we evaluated several methods aimed at mitigating overthinking, yet the influence of internal bias persisted under all conditions.

## 1 INTRODUCTION

Current o1/R1-type reasoning models (OpenAI et al., 2024; DeepSeek-AI et al., 2025a) have demonstrated outstanding performance with their ability to spontaneously reflect and correct errors (Xu et al., 2025a). However, the reasoning models tend to overthink (Chen et al., 2025b), which is characterized by behavioral patterns such as repeatedly reaching the same conclusion without contributing to the final answer. This redundant thinking results in significant waste in computation.

Despite efforts in reducing overthinking through training (Chen et al., 2025b; Su et al., 2025; Team et al., 2025; Shen et al., 2025; Arora and Zanette, 2025) or manual intervention during decoding (Chen et al., 2025a; Zhang et al., 2025a; Ma et al., 2025), an important question remains underexplored: *what drives the reasoning models to exhibit such a overthinking tendency ?*

We identify the internal bias of reasoning models as one important reason for their overthinking. Our hypothesis is twofold: first, upon encountering a problem, the model forms a preliminary and intuitive guess, before engaging in formal reasoning; second, when this initial guess conflicts with the outcome of deliberate reasoning, the model is more likely to enter a state of excessive reflection. We refer to this guess as an **internal bias**, to emphasize that it may not be explicitly output by the model, and is not derived from rigorous reasoning but originates solely from the input question. This entire process is illustrated in Figure 1 [1].

Extensive statistical experiments across different model families (DeepSeek (DeepSeek-AI et al., 2025b), Qwen (Qwen et al., 2025)), model sizes (14B, 32B and 671B), downstream tasks (character operations, logical reasoning and mathematical reasoning) and questioning languages (English and Chinese) consistently demonstrate that internal bias has a widespread influence on reasoning behavior. Specifically, the greater a model's internal bias deviates from its reasoning result, the more

---

[†]Corresponding author.
[1]All responses in these examples are generated by DeepSeek-R1 (2025/01/20). The complete model responses and details of internal bias detection can be found in Appendix A.1 and A.2.

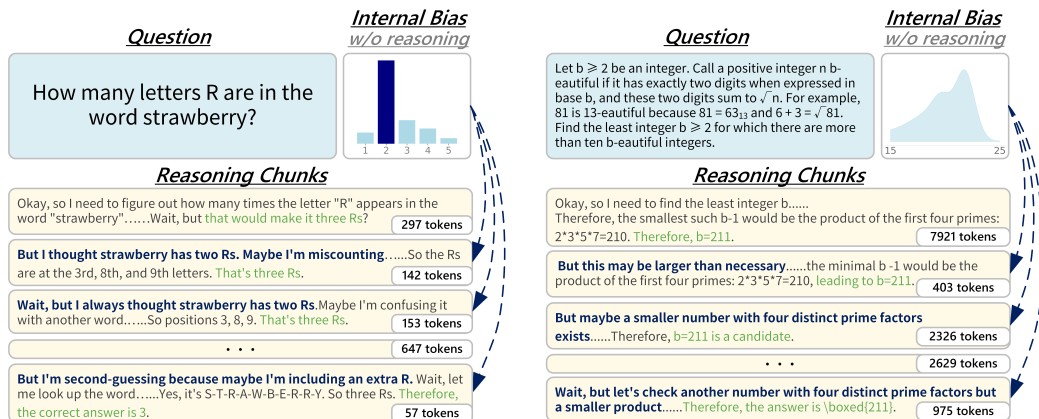

Figure 1: Two examples illustrating the existence of internal bias. Green texts denote correct answers derived from reasoning, while the navy-colored portions show the influence of internal bias. In the left example (a simpler case), the model develops an internal bias of "2", conflicting with the reasoning result "3". In the more complex example (from AIME 2024) on the right, the model predicts the answer to be approximately "20", significantly deviating from the correct value "211" obtained via reasoning. For both examples, the reasoning process are manually separated into chunks for better illustration, where the model obtains the correct answer in the first chunk, but internal bias still triggers later reflection. The number in the bottom-right corner of each chunk indicates the length of it. It is clear that the model spends much more tokens in reflection due to internal bias.

likely it is to engage in excessive reflection. In extreme cases, this tendency can even lead to the parroting behavior (Xu et al., 2022).

To establish a causal link between internal bias and overthinking in reasoning models, we propose two counterfactual validations: a reasoning trajectory intervention and a bias injection. The first intervention removes the input question after the model generates an answer during its reasoning steps, forcing it to decide whether to continue reasoning based solely on its own past reasoning steps, thereby preventing the reactivation of input-dependent biases. This intervention reduces redundant reasoning length by 31% to 53% across both synthetic simple tasks and complex logical and mathematical reasoning benchmarks, while largely maintaining or even improving accuracy, indicating that the reduced reasoning steps are largely redundant. The second intervention, bias injection, deliberately manipulates the model's internal bias through controlled training signals, demonstrating that erroneous biases exacerbate overthinking, whereas accurate ones suppress it.

We further conducted attention analysis along the reasoning process, to further understand how internal bias affects reasoning. Interestingly, we found that after completing a reasoning step, the model tends to excessively focus (attention) on the question description, which may introduce its internal bias into the decision-making process on whether further reflection is needed. This suggests that models might implicitly compare their reasoning results with its internal bias, leading to extra reasoning in some cases despite having already arrived at the correct answer multiple times.

Finally, we tested several existing methods for mitigating overthinking. We found that, despite reducing average reasoning length, these methods fail to eliminate the influence of internal bias, demonstrating its resilience in model reasoning.

Our contributions are as follows:

- Identifying internal bias as one important reason for overthinking in reasoning models, and validate the universality of this phenomenon across various experiment settings.

- Demonstrating the causal relationship between internal bias and overthinking with rigorous experimental evidence.

- Discovering that internal bias influences the reasoning process through the model's excessive attention on the input question.

- Testing several methods of mitigating overthinking, and revealing they are ineffective in eliminating the influence of internal bias.

## 2 RELATED WORK

### 2.1 MITIGATING OVERTHINKING IN REASONING MODELS

With the emergence of DeepSeek-R1 (DeepSeek-AI et al., 2025a) and OpenAI-o1 (OpenAI et al., 2024) style reasoning models, the overthinking phenomenon, in which models produce excessively long and unnecessary chains of thought, has become a widely recognized issue. This has spurred growing interest in efficient reasoning approaches aimed at mitigating such behavior (Yue et al., 2025; Sui et al., 2025), which can be broadly categorized into three families: training-time, inference-time and prompting approaches.

*Training-time* approaches constrain reasoning length by encouraging shorter reasoning chains (Chen et al., 2025b; Yu et al., 2025; Cheng et al., 2025), or by incorporating difficulty-dependent length penalties that allow the model to adaptively select reasoning paths (Shen et al., 2025; Liu et al., 2025; Luo et al., 2025a). *Inference-time* approaches reduce redundant thinking through decoding interventions, such as confidence- or consistency-based dynamic early exit Zhang et al. (2025a); Yang et al. (2025) or representation-level steering of model behavior (Chen et al., 2025a; Huang et al., 2025; Eisenstadt et al., 2025). *Prompting-based* approaches control reasoning overhead through input design, for example by imposing token budgets (Han et al., 2025), using draft-style prompting (Xu et al., 2025b), or even explicitly skipping reasoning when appropriate (Ma et al., 2025).

These methods aim to control model behavior without investigating the root cause of overthinking. Our work addresses this gap by showing that internal bias is one of the key underlying factors driving overthinking.

### 2.2 BIAS AND UNFAITHFULNESS IN LANGUAGE MODELS

Language models often develop various biases or priors. One prominent category involves systematic biases in sensitive domains such as law and social ethics (Gallegos et al., 2024), which may raise fairness concerns in practical applications. More broadly, models exhibit a wide range of behavioral biases, including a tendency to produce the most "common" rather than the correct answers for certain question types (Zhang et al., 2025b), susceptibility to irrelevant contextual cues or format-specific patterns (Minder et al., 2025; Weston and Sukhbaatar, 2023), and preference priors induced inevitably by reinforcement learning reward (Greenblatt et al., 2024; Denison et al., 2024). Ameisen et al. (2025) found that language models may have separate neural pathways that lead to a rough estimate of the answer when solving simple addition tasks, providing additional mechanistic evidence for the presence of such biases. The internal bias discussed in this paper is closely related to this type of bias, and we focus on how such direct estimations of the answer can trigger overthinking.

Unfaithfulness is closely intertwined with biases. When a model produces Chain-of-Thought (CoT), the generated steps may not necessarily reflect the model's true reasoning process. Paul et al. (2024) shows that interfering with the CoT sentences typically has little effect on its final answer, suggesting a spurious reasoning phenomenon during reasoning. Turpin et al. (2023) and Chen et al. (2025c) demonstrate that models often produce explanations for a pre-selected answer. Arcuschin et al. (2025) extend this line of evidence by identifying bias-driven implicit post-hoc rationalization: systematic biases shape the model's internal decision, and explicit CoT is then used to rationalize the resulting biased answer, producing a divergence between surface-level reasoning and underlying computation even under natural, non-adversarial prompts. These unfaithfulness works involve models deviating from normal reasoning processes. However, our work finds that even when models successfully follow correct reasoning procedures, they can still be influenced by bias at critical reflection steps, leading to overthinking.

## 3 MEASURING INTERNAL BIAS IN REASONING MODELS

We now introduce several definitions that will form the basis for quantifying and analyzing internal bias in the later sections.

**Direct Answer** [2] As introduced in § 1, we conceptualize the model's *internal bias* as its preliminary guess to a question, formally denoted as $a_{\text{bias}} = f(q; \theta)$, where $q$ is the input question, $\theta$ represents the model parameters and $a_{\text{bias}}$ denotes the biased answer. To observe this implicit guess, we force the model to skip reasoning and immediately output an answer. We name this answer as *direct answer*. Specifically, we use templates containing a no-reasoning prompt followed by the special token `</think>` to indicate termination of thought. The content generated after this prompt reflects the preliminary judgment of the model formed prior to detailed reasoning. An example of such a template is provided below. In contrast, the final answer after full reasoning is denoted as $a_{\text{final}}$.

```
<|User|>{question}<|Assistant|><think>
Let me answer him without thinking more.</think>
Answer:
```

**Internal Bias as a Distribution** As in a probabilistic generation process, the generated direct answer may also be affected by the sampling strategy, e.g. the prompt or the temperature. On the other hand, the model's internal bias may also exist as a distribution over possible answers rather than a single deterministic guess. So we use $\tilde{a}_{\text{bias}}$ to denote the internal bias distribution. To approximate this distribution, we adopt a multi-sampling approach with different templates. Empirically, we collect 64 direct answers for each question in our experiments, which is a better approximation of the model's internal bias by observing its behavior under varied conditions. All templates and decoding details can be found in Appendix B.

**Deviation Degree** The internal bias of a model may deviate from its own reasoning process, and different deviation degree may have varying effects on its behavior. We define the bias deviation degree $D_{\text{bias}} = \text{dist}(\tilde{a}_{\text{bias}}, a_{\text{final}})$, where $\text{dist}(\cdot, \cdot)$ is a task-specific distance function. For tasks with **numerical answers**, we compute the *mean absolute error (MAE)* between the direct answers and $a_{\text{final}}$. For tasks with **categorical answers**, such as multiple-choice questions, we approximate $D_{\text{bias}}$ using *inconsistency rate*, defined as the proportion of direct answers that differ from $a_{\text{final}}$. These serve as approximate quantitative estimates of the gap between the model's direct guesses and the reasoning output.

## 4 IMPACT OF INTERNAL BIAS ON THE REASONING PROCESS

In this section, we take a macroscopic view of the impact of internal bias on reasoning models. Our analysis includes the examination on the direct answer and its relation to reasoning length, as well as the understanding of the parroting phenomenon.

### 4.1 SETUPS

**Tasks.** We design a controllable character manipulation dataset: **CharCount (zh)** and **CharCount (en)**: The model is tasked with counting how many specific letters are in a given word. To account for linguistic variation, the dataset is split into Chinese and English subsets, based on prompting language. More details are presented in Appendix C. We also utilize several open-source reasoning datasets: **KnowLogic** (Zhan et al., 2025): A complex dataset synthesized using real-world knowledge and logical reasoning rules, where each question provides four possible answers A, B, C, and D, and more than one may be correct. **AIME 2024** and **AIME 2025** [3]: Challenging mathematical reasoning datasets.

**Models.** To demonstrate the universality of internal bias, we conduct experiments across different model families and sizes. Specifically, we select DeepSeek-R1 (DeepSeek-AI et al., 2025a), R1-distill-Qwen-14B, and QwQ-32B (QwenTeam, 2025).

---

[2]In Appendix J, we verify that direct answers are consistent with latent representations, demonstrating that they can fairly and accurately capture the internal bias.

[3]`https://artofproblemsolving.com/wiki/index.php/AIME_Problems_and_Solutions`

**Experiment Details.** For each question, we let the model to generate 64 direct answers and one normally reasoned response. Similar to many other works (Chen et al., 2025a; Zhang et al., 2025a), we count reflection-related keywords (e.g. "wait") to roughly illustrate the number of the model's reflections. The full keyword lists are in Appendix D. All experiments were conducted on NVIDIA RTX A6000 GPUs.

## 4.2 OVERALL RESULTS

Table 1: Results on **CharCount (zh)**, **KnowLogic** and **AIME 2024** datasets.

| **CharCount (zh)** | Acc | $Acc_{direct}$ | $L_{low}$ | $L_{high}$ | $R_\Delta$ |
|---|---|---|---|---|---|
| DeepSeek-R1 | 99.2% | 55.8% | 556.7 | 735.6 | 32.1% |
| QwQ-32B | 92.6% | 36.8% | 715.6 | 866.0 | 21.0% |
| R1-Distill-Qwen-14B | 73.3% | 17.3% | 944.4 | 1228.7 | 30.1% |
| **KnowLogic** | Acc | $Acc_{direct}$ | $L_{low}$ | $L_{high}$ | $R_\Delta$ |
| DeepSeek-R1 | 54.0% | 29.4% | 2965.7 | 4214.3 | 42.1% |
| QwQ-32B | 51.6% | 29.7% | 5694.7 | 7167.1 | 25.9% |
| R1-Distill-Qwen-14B | 27.2% | 24.9% | 5713.8 | 6927.6 | 21.2% |
| **AIME 2024** | Acc | $Acc_{direct}$ | $L_{Low}$ | $L_{High}$ | $R_\Delta$ |
| DeepSeek-R1 | 76.7% | 3.3% | 7934.8 | 9764.2 | 23.6% |
| QwQ-32B | 73.3% | 0.0% | 10239.7 | 13521.6 | 32.1% |
| R1-Distill-Qwen-14B | 63.3% | 0.0% | 8882.1 | 12709.9 | 43.1% |

Table 1 presents results on CharCount (zh), KnowLogic and AIME 2024 datasets, with the rest results provided in Appendix F.1. In each table, "Acc" is the accuracy after reasoning; "$Acc_{direct}$" is the accuracy of direct answers computed by majority votes; $L_{low}$ and $L_{high}$ are the average reasoning length (number of tokens) for cases with lower and higher half of deviation degree; $R_\Delta$ represents the relative length increase of the high-deviation group compared with the low-deviation group.

**Direct answer's accuracy is relatively low.** The model exhibits a high error rate in its direct answers, indicating that it often fails to perform a correct guess, which is why we define such answers as internal *bias*. The result also suggests that bias is largely detrimental in most cases, especially as the complexity of the questions increases.

**The reasoning length of the high-deviation group is longer than that of the low-deviation group.** The increase in length is at least 21.0%, and reaches 42.1% in some cases. This observation intuitively demonstrates that the degree of internal bias deviation has a significant impact on the model's reasoning length. The trends are consistent across all models and datasets.

## 4.3 FINE-GRAINED ANALYSES

To conduct a more fine-grained analysis of the impact of internal bias, we further divide the deviation degree into four smaller intervals. Figure 2 shows the results of R1-distill-Qwen-14B on CharCount (zh), representing tasks with the numerical answers, and QwQ-32B on KnowLogic, representing tasks with the categorical answers. More results are presented in Appendix F.2.

Reasoning models may naturally engage in reflection, so even for cases with low deviation bias (MAE< 0.5 or Inconsistency Rate < 25%), reflective behavior can still be observed following the initial reasoning step. The main reason for this reflection may be to check whether the result is correct. For these cases, the number of reflection keywords is usually small.

**The greater the deviation of the internal bias is, the more severely the model overthinks.** As internal bias deviation increases, both the average output length and the count of reflection keywords rise sharply, which aligns with the earlier observation that internal bias triggers overthinking. For these thinking processes, the model tend to generate much more reflection keywords, because it is switching between different thoughts. These steps do not involve a thorough examination of the reasoning process but instead result primarily in increased token consumption without meaningful self-evaluation. This trend is consistently observed across all experiments.

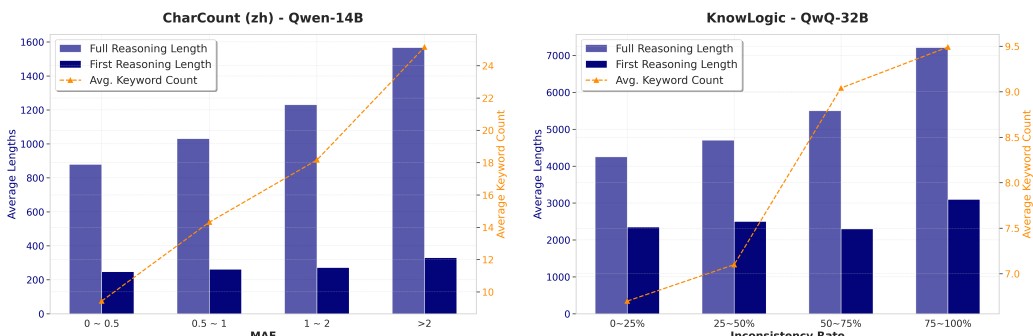

Figure 2: Correlation between deviation degree and reasoning behavior. The light-colored bars represent the full reasoning length, while the dark-colored bars indicate the position at which the model first provides an answer. The orange line shows the number of reflection keywords. Qwen-14B here is short for R1-distill-Qwen-14B. Appendix E describes the method used to identify the position at which the model first provides an answer during its reasoning process.

**Internal bias rather than question complexity causes the observed trends.** One might think that more difficult questions need longer thinking. Thus we collect the first reasoning length of each question, i.e. the length of the first complete thought that leads to an answer, which could be seen as an indicator of the complexity of the problem. As shown in Figure 2, the first reasoning lengths are almost the same for examples within the same task, indicating that the complexity of questions are almost the same in different groups in the same task. Therefore, internal bias is more likely to be the reason for overthinking.

**In extreme cases, internal bias can lead to a parroting phenomenon.** "Parroting" refers to situations in which the model repeatedly generates exactly the same content, thus fails to provide a final answer. Taking R1-Distill-Qwen-14B performing CharCount (zh) as an example, among all parroting cases, the direct answer accuracy is only 4.4%, compared to 17.3% across all cases (Table 1). Conversely, if we estimate MAE between the last numeric value in the model's output and direct answers, and split all cases into two groups based on MAE, the high-MAE group exhibits a parroting probability of 6.6%, while the low-MAE group shows a significantly lower rate of only 2.8%, suggesting a correlation between deviation degree and the parroting behavior. This highlights the underlying reason behind the parroting phenomenon: driven by strong internal bias, the model engages in excessive reflection early in the reasoning process, which further reinforces its tendency to reflect. This self-perpetuating cycle ultimately leads to repetitive output during the later stages.

## 5 CAUSAL EFFECTS OBSERVED THROUGH COUNTERFACTUAL INTERVENTIONS

In this section, we employ two distinct counterfactual intervention methods to establish the causal relationship between internal bias and overthinking in reasoning models. We use R1-Distill-Qwen-14B model for controllable analysis.

### 5.1 REMOVING QUESTION FROM PROMPT

When the model first generates an answer during its reasoning steps, we immediately remove the input question from the prompt and allow the model to continue generating without access to the original task. Since a complete reasoning process has already incorporated the necessary information from the question, this forces the model to decide solely based on its own prior reasoning trajectory whether to engage in further reflection or terminate the thinking process. Under this intervention, we examine whether the model reduces its reasoning length and assess how its final performance on answering the question is affected.

We validate this on the full KnowLogic, AIME 2024, and AIME 2025 datasets, as well as random selected 1,000-example subsets from CharCount (en) and CharCount (zh). We introduce the length

reduction ratio $r = (L_{ori} - L_{rem})/(L_{ori} - P_{first})$ to measure the length reduction after the first reasoning answer is obtained, where L denotes the output length, the subscript ori/rem denotes the original and question-removed situation, and $P_{first}$ denotes the position where the first reasoning answer is obtained, i.e. where we start to remove the input question. Results are shown in Table 2.

Table 2: Results of the removing question intervention.

| Dataset | $Acc_{ori}/Acc_{rem}$ | $L_{ori}/L_{rem}$ | $P_{first}$ | $r$ |
|---|---|---|---|---|
| CharCount (en) | **93.8** / 93.2 | 541.0 / 453.3 | 258.8 | 31.1 |
| CharCount (zh) | **73.4** / 72.9 | 1081.9 / 782.5 | 245.1 | 35.8 |
| KnowLogic | 27.2 / **29.3** | 6320.7 / 5857.5 | 4921.8 | 33.1 |
| AIME 2024 | 63.3 / **66.7** | 10796.0 / 7706.1 | 5022.7 | 53.5 |
| AIME 2025 | 36.7 / **46.7** | 13428.9 / 10303.0 | 4323.4 | 34.3 |

**Removing question consistently leads to a reduction in redundant reasoning.** Across all datasets, removing the question leads to a reduction ratio $r$ ranging from 31.1% to 53.5%, causally confirming that the model's excessive focus on the question section is a key driver of overthinking.

**Removing question leads to better performance in complex tasks.** We observe performance improvements on complex datasets. Case studies reveal that these improvements are largely attributed to previously parroting behaviors now being resolved correctly, which, as discussed earlier in § 4.3, are also strongly linked to the influence of internal bias. The observed accuracy decrease in simpler tasks is slight and likely stems from the short reasoning chains, which may be more easily disrupted by our coarse-grained intervention. These indicate that the reduced reasoning length primarily consists of redundant thoughts that do not contribute to the model's final answer.

## 5.2 BIAS INJECTION

We further conduct a bias injection experiment, deliberately manipulating the model's internal bias toward specific questions and examine whether its reasoning behavior changes accordingly.

Specifically, we select 500 samples each with the the lowest and highest deviation degree ($D_{bias}$) from the CharCount (zh) dataset. Based on this, we design a bias injection method: we construct 50 rephrased declarative statements for each sample and then fine-tune the model on these statements to perform a sample-wise bias injection, and then test on the exact same sample. We design two training regimes: a) *Low2Wrong*: construct wrong declarative statements with the lowest $D_{bias}$ samples; b) *High2Correct*: construct correct statements with the highest $D_{bias}$ samples; c) *Random2Correct*: test on high $D_{bias}$ samples, but construct random correct statements that differ from the testing sample; d) *Random2Wrong*: test on low $D_{bias}$ samples, but construct random wrong statements. The last two serve as baselines to rule out influence of the bias injection training itself. An example of the declarative statement, additional data and training details are provided in Appendix H.

As shown in Table 3, when comparing each setting with its corresponding *Random* baseline, we observe that in the *Low2Wrong* setting, the reasoning length increases from 355.5 to 454.9, whereas in the *High2Correct* setting it decreases from 600.8 to 412.1. These changes indicate that injecting incorrect biases makes the model more prone to reflection, while injecting correct ones reduces unnecessary deliberation, providing strong evidence for a causal relationship between the internal bias and the model's reflective reasoning patterns. This may indicate that the model's incorrect internal knowledge distribution is one of the origins of internal bias.

Table 3: Bias injection results on subsampled CharCount(zh) dataset.

| Setting | Length | Acc |
|---|---|---|
| *Random2Wrong* | 355.5 | 89.5% |
| *Low2Wrong* | 454.9 | 83.4% |
| *Random2Correct* | 600.8 | 67.0% |
| *High2Correct* | 412.1 | 76.6% |

## 6 INTERPRETABILITY ANALYSIS

As shown in Figure 1, although with the existence of internal bias, LLMs can indeed successfully derive the correct answer with its own reasoning. An interesting question is: *why the model keeps the internal bias in mind even after long steps of reasoning ?* We observe that the model pays **excessive attention** to the input question when deciding whether to engage in further reflection. Intuitively, the model may be referring the question to decide whether the current reasoning is correct. But this heightened attention also reactivates the internal bias, influencing the decision to reflect. In Appendix J.1, we conduct a probing study that verifies the hidden states from the question segment are highly correlated with the direct answers and thus encode the internal bias. This supports that bias could be reactivated through attention to the input. We now examine attention behavior through both an illustrative example and quantitative analysis.

### 6.1 THE "STRAWBERRY" EXAMPLE

< | begin_of_sentence | > < | User | > How many letters ' r ' are there in word ' str aw berry '? Answer directly with an Arabic number . < | Assistant | > <think>

Okay , so I need to figure out how many times the letter ' r ' appears in the word ' str aw berry '. Let me start by writing out the word to visualize it better : S -T -R -A -W -B -E -R -R -Y . Wait , no , that 's not quite right . Let me make sure I spell it correctly . S -T -R -A -W -B -E -R -R -Y ? Hmm , actually , I think I might be adding an extra ' r ' there . Let me check again .

The word is ' str aw berry '. Breaking it down letter by letter : S , T , R , A , W , B , E , R , R , Y . Wait , that 's 1 0 letters . But I 'm trying to count the ' r 's . So starting from the beginning : S ( no ), T ( no ), R ( yes , that 's one ), A ( no ), W ( no ), B ( no ), E ( no ), R ( that 's two ), R ( that 's three ), Y ( no ). So I count **three** 'r's.

**Wait**, but I think I might be making a mistake here...

Figure 3: The "strawberry" example as an illustration of abnormally high attention scores on question part when a reflection token is about to be output. Color intensity is used to represent the ratio of attention scores assigned to each preceding tokens at the following two steps: generating "three" and generating "Wait". Darker red indicates a higher relative attention at the reflection point when "Wait" is generated, while darker blue reflects higher relative attention when generating "three".

As a direct illustration, we revisit the "strawberry" example from Figure 1. We compare the attention score for the context tokens at two different steps during the reasoning: the answer point, where the model generates an intermediate result; and the reflection point, where the model decides whether to perform a further reflection.

The visualization in Figure 3 reveals a clear shift in attention at the reflection point: Compared to the the answer point, the model focuses more heavily on the question part (the first segment/paragraph), its own paraphrasing of the question, and earlier superficial reasoning steps with reflection tokens (the second segment), while reducing its attention to the core reasoning process that actually obtains the answer (the third segment) when deciding whether a reflection token should be output or not.

### 6.2 STATISTICAL RESULTS ON CHARCOUNT

To further confirm the model's excessive focus on the input question, we conduct a statistical analysis. We categorize all preceding tokens into three distinct categories: (1) *Question*, which refers to the tokens in the input question. (2) *Mid_Results*, which corresponds to tokens representing intermediate counting results, e.g. "3" or "three". (3) *Others*. For a token at position $p$, we can compute the *average normalized score* $s_p^c$ it assigns to previous tokens of category $c$, and further aggregate them across groups $G \in \{\text{Reflection}, \text{Other}\}$:

$$s_p^c = \frac{1}{N_c} \sum_{i<p, i\in c} \frac{a_p^i}{1/p}, \quad S_G^c = \frac{1}{|G|} \sum_{p\in G} s_p^c$$

where $N_c$ is the number of tokens that belongs to category $c$ preceding to token $p$, and $a_p^i$ is the attention weight from token $p$ to $i$. Dividing the by $1/p$ normalizes for positional imbalances, as it corresponds to the uniform attention baseline, enabling fair comparison across different token positions. The group-level score $S_G^c$ quantifies the average extent to which tokens in group $G$ attend to category $c$. All calculations exclude the first three tokens of the sequences to exclude the attention sink phenomenon (Xiao et al., 2024).

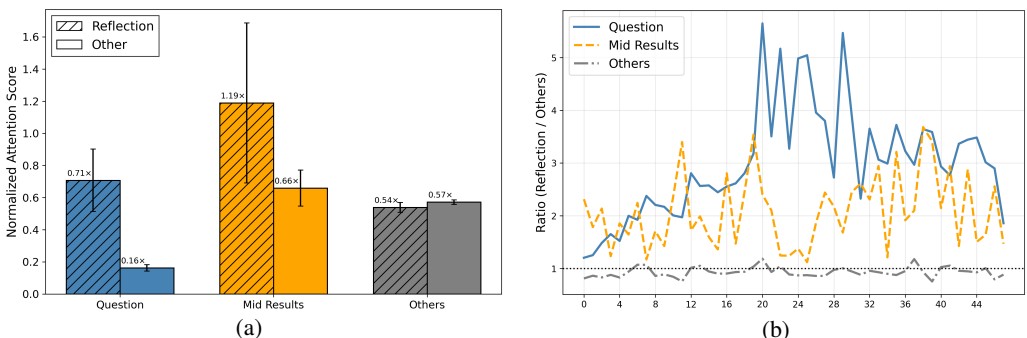

Figure 4: (a) Group-level scores $S_G^c$ with averaged attention scores from layers 21 to 30. Similar visualizations for other layers are in Appendix G and the trends are the same. (b) The ratio of $S_{\text{Reflection}}^c / S_{\text{Other}}^c$ across all layers.

Figure 4(a) shows the model generally pays low attention to the question section during most of the reasoning process, while upon reflections it exhibits a sharp increase to more than four times its original level. Although attention to previous intermediate results also increases, the proportion of attention allocated to the question section still rises significantly. Figure 4(b) shows the layer-wise ratio of group-level scores $S_G^c$ for tokens in reflection steps versus other tokens. A sharp increase in attention to question tokens occurs in the middle-to-later layers, where the model engages in active reasoning (Zhao et al., 2024; Wendler et al., 2024). These attention patterns may reactivate internal bias during the model's decision on whether to initiate further reflection and lead to overthinking.

## 7 MITIGATION TRIALS

Finally, we apply some existing techniques for mitigating overthinking and evaluate whether they can also reduce the influence of internal bias.

We reproduce two mainstream categories of approaches: (1) training-time reasoning optimization, exemplified by First-Correct Solutions with Reflection (denoted as FCS) (Chen et al., 2025b), which focuses on constructing high-quality short reasoning data and refining the model through post-training via Supervised Fine-Tuning (SFT) or Direct Preference Optimization (DPO); and (2) inference-time reasoning interventions, including SEAL (Chen et al., 2025a), which steers the reasoning trajectory by intervening hidden states during decoding, and PROBE (Zhang et al., 2025a), which probes internal representations and determines an appropriate time to manully perform early stopping. We implement these methods on R1-Distill-Qwen-14B model (denoted as Qwen-14B) and the 1000-sample subset of CharCount (zh) dataset; and additionally reproduce the best performed method FCS$_{\text{SFT}}$ on AIME 2024. The implement details are in Appendix I. The results are summarized in Table 4, where the symbols follow the same convention as in Table 1, and include a "Remove" entry corresponding to the question-removal intervention (§ 5.1) as a reference.

On CharCount dataset, existing methods perform well in shortening reasoning trajectories, yet show minimal effectiveness in mitigating internal bias, and in some cases even exacerbating

its influence, as evidenced by the comparable or even larger $R_\Delta$. As a reference, question-removal can achieve an $R_\Delta$ of only 15.2%. The training data for these methods are drawn from the CharCount dataset itself, which may contribute to the observed performance improvement. Notably, the simplest SFT method achieves the highest accuracy and best $R_\Delta$ among existing methods. However, on the more complex task AIME 2024, it suffers a significant drop in accuracy, indicating that shortened reasoning chains may impair the model's reasoning ability for complex problems, which aligns with the findings of Chen et al. (2025b). These results suggest that existing methods achieve shorter reasoning traces only superficially, some-

Table 4: Mitigation trials results in CharCount and FCS$_{\text{SFT}}$ results in AIME2024.

| **CharCount** | Acc | $L_{\text{low}}$ | $L_{\text{high}}$ | $R_\Delta$ |
|---|---|---|---|---|
| Qwen-14B | 73.4% | 934.7 | 1229.1 | 31.5% |
| + Remove | 72.9% | 727.3 | 837.8 | 15.2% |
| + FCS$_{\text{DPO}}$ | 76.7% | 555.3 | 812.8 | 46.3% |
| + FCS$_{\text{SFT}}$ | 78.9% | 451.3 | 572.0 | 26.7% |
| + SEAL | 77.4% | 581.3 | 805.1 | 38.5% |
| + PROBE | 73.1% | 702.6 | 912.5 | 29.9% |
| **AIME2024** | Acc | $L_{\text{low}}$ | $L_{\text{high}}$ | $R_\Delta$ |
| Qwen-14B | 63.3% | 8882.1 | 12709.9 | 43.1% |
| + Remove | 66.7% | 7252.6 | 8159.6 | 12.5% |
| + FCS$_{\text{SFT}}$ | 50.0% | 7204.6 | 9718.1 | 34.9% |

times even harming the model's reasoning capability. Crucially, the underlying reflection pattern driven by internal bias continues to profoundly affect reasoning efficiency.

## 8 CONCLUSION AND DISCUSSION

We identify internal bias as a key reason for overthinking in reasoning models, and demonstrate this relevance across different experimental settings. Conterfactual bias intervention and bias injection results provide rigorous evidence of the causal link between internal bias and overthinking. Further interpretability analyses reveal that excessive attention to the question during reflection likely reactivates the internal bias, leading to redundant reflections even after long reasoning chains. Finally, we find that existing overthinking mitigation methods may fail to eliminate the influence of internal bias, highlighting its resilience and the importance of addressing this issue. Our findings call for a shift in how we understand reasoning models and open new directions for building more efficient, self-aware, and adaptive reasoning systems.

Nevertheless, several aspects still merit deeper discussion, including the origins of internal bias, the dynamics of its influence, potential mitigation strategies, and the limitations of this work.

**Origins of Internal Bias**   Internal biases may arise from multiple interacting factors rooted in training methods and data. For instance, reinforcement learning with verifiable rewards only may implicitly promote repeated self-checking, while inaccurate knowledge distributions can create conflicts between the model's internal states and the correct reasoning path, triggering reflections, which has already received preliminary verification in the bias injection experiments in §5.2. Further analysis of knowledge storage mechanisms like FFN neurons may provide additional insights into biases.

**Transition from Bias to Reasoned Output**   We are interested in when a model becomes "convinced" by the reasoning outputs and discards its internal bias. We hypothesize that this transition is gradual: as reasoning progresses, attention to the question becomes diluted, decreasing the influence of internal bias and allowing trust in the ongoing reasoning trajectory to dominate. Supplementary results supporting this hypothesis are provided in Appendix K.

**Potential Solutions**   Since bias is deeply rooted within the model, we argue that the most efficient approach is not to eliminate bias through training, but rather to decouple its influence during inference. Inspired by the analysis in §6, an attention-based early-exit mechanism may offer a promising direction. We explored this idea in Appendix I.3 and observed improvement on the CharCount (zh) dataset. However, developing more robust solutions remains future work.

**Limitations**   All analyses in this paper are restricted to tasks with outputs that are explicit numerical values, ranges, or multiple-choice options. We acknowledge that for open-ended tasks, bias is not yet well defined, as it may manifest in more complex forms.

ACKNOWLEDGEMENT

We would like to thank the anonymous reviewers for their insightful comments. Shujian Huang is the corresponding author. This work is supported by National Science Foundation of China (No. 62376116), research project of Nanjing University-China Mobile Joint Institute (NJ20250038), the Fundamental Research Funds for the Central Universities (No. 2024300507), Fundamental and Interdisciplinary Disciplines Breakthrough Plan of the Ministry of Education of China (No. JYB2025XDXM118).

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

# A    SUPPLEMENTS FOR FIGURE 1

## A.1    FULL ANSWERS

Below are the complete responses corresponding to the examples in Figure 1. The "<think>" and "</think>" tokens are added for clearer demonstration. And we mark thoughts in the reasoning process that are explicitly affected by the internal bias in bold font.

---

*QUESTION*:
How many letters R are in the word strawberry?

*DEEPSEEK-R1*:
<think>
Okay, so I need to figure out how many times the letter "R" appears in the word "strawberry." Let me start by writing down the word so I can look at each letter one by one.
The word is S-T-R-A-W-B-E-R-R-Y. Let me count each letter carefully.
Starting with the first letter: S. That's an S, not an R. Next is T. Still not an R. The third letter is R. Okay, that's one. Then A, W, B, E—those aren't R. Then comes another R. So that's two. Next is another R. Wait, let me check again. The word is strawberry. Let me spell it out: S, T, R, A, W, B, E, R, R, Y. Hmm, after E, there are two Rs and then a Y. So after E, it's R, R, Y. So that would be two Rs after the E. Wait, so in total, the Rs are at the third position, then the eighth and ninth positions? Let me count again to be sure.
Breaking it down:
1. S
2. T
3. R (1)
4. A
5. W
6. B
7. E
8. R (2)
9. R (3)
10. Y
Wait, but that would make it three Rs? **But I thought strawberry has two Rs**. Maybe I'm miscounting. Let me check the spelling again. Strawberry is spelled S-T-R-A-W-B-E-R-R-Y. So after B, it's E, then R, R, Y. So positions 8 and 9 are both Rs. So that's two Rs there, plus the one at position 3. So total of three Rs? **But I feel like that's not right**. Let me verify by writing the word and underlining each R.
S T R A W B E R R Y
So the Rs are at the 3rd, 8th, and 9th letters. That's three Rs. **Wait, but I always thought strawberry has two Rs.** Maybe I'm confusing it with another word. Let me check the correct spelling. **Maybe I added an extra R by mistake.** Let me look up the spelling of strawberry to confirm. [Pauses to recall] No, strawberry is S-T-R-A-W-B-E-R-R-Y. So yes, two Rs at the end before the Y. So that's three Rs in total? Wait, no. Let me count again: S (1), T (2), R (3), A (4), W (5), B (6), E (7), R (8), R (9), Y (10). So positions 3, 8, 9. That's three Rs. **But I think I might be making a mistake here because when I say "strawberry" out loud, it sounds like two Rs.** Let me check another way. Maybe I'm breaking down the syllables wrong. Straw-ber-ry. Wait, the "ber" part is B-E-R, then "ry" is R-Y. So maybe in the spelling, after B-E-R, there's another R before Y. So that's two Rs in the last part. So total of three? **But I'm confused now**. Let me write it again:
S T R A W B E R R Y
So letters: 1.S, 2.T, 3.R, 4.A, 5.W, 6.B, 7.E, 8.R, 9.R, 10.Y. So indeed, three Rs. **But I must be wrong because I remember strawberry having two Rs**. Wait, maybe the correct spelling is S-T-R-A-W-B-E-R-Y, with only one R after E. Let me confirm. No, no, I think it's definitely spelled with two Rs at the end. Because the word comes from "straw" + "berry," and "berry" is B-E-R-R-Y. So "berry" has two Rs, right? Yes, berry is B-E-R-R-Y. So strawberry is straw + berry, so combining them would be S-T-R-A-W-B-E-R-R-Y. Therefore, three Rs: one in "straw" (the third letter) and two in "berry" (the eighth and ninth letters). So total of three Rs. **Wait, but that contradicts my initial thought**. Let me check a dictionary or something. [Pretends to check a dictionary] Hmm, according to the dictionary, strawberry is spelled S-T-R-A-W-B-E-R-R-Y. So yes, three Rs. Therefore, the answer should be three. **But wait, when I say the word, it's pronounced with two Rs? Or maybe I'm just miscounting**. Let me try pronouncing it: straw-ber-ry. The "ber" is one R, and the "ry" is another R? No, actually, "berry" is pronounced with two Rs: B-E-R-R-Y. So in the spelling, it's two Rs, making the total in

strawberry as three. Wait, no. Wait, "straw" is S-T-R-A-W, which has one R. Then "berry" is B-E-R-R-Y, which has two Rs. So combined, that's three Rs. Therefore, the answer is three. **But I need to make sure I'm not adding an extra R**. Let me count again letter by letter:

S (1) - no
T (2) - no
R (3) - 1
A (4) - no
W (5) - no
B (6) - no
E (7) - no
R (8) - 2
R (9) - 3
Y (10) - no

Yes, three Rs. So the answer is 3. **But I'm second-guessing because maybe I'm including an extra R**. Wait, let me look up the word strawberry's spelling again mentally. Yes, it's S-T-R-A-W-B-E-R-R-Y. So three Rs. Therefore, the correct answer is 3.
</think>

To determine how many times the letter "R" appears in the word "strawberry," let's break it down step by step:

Spell the word: S-T-R-A-W-B-E-R-R-Y.

Identify each letter's position:

1: S
2: T
3: R
4: A
5: W
6: B
7: E
8: R
9: R
10: Y

Count the occurrences of "R":

Position 3: R (1)
Position 8: R (2)
Position 9: R (3)

Final Answer: The word "strawberry" contains 3 letters "R".

---

*QUESTION*:

Let $b \geq 2$ be an integer. Call a positive integer $n$ *b-eautiful* if it has exactly two digits when expressed in base $b$, and these two digits sum to $\sqrt{n}$. For example, 81 is 13-eautiful because $81 = \underline{63}_{13}$ and $6 + 3 = \sqrt{81}$. Find the least integer $b \geq 2$ for which there are more than ten $b$-eautiful integers.

*DEEPSEEK-R1*:

<think>
Okay, so I need to find the least integer b (where b is at least 2) such that there are more than ten b-eautiful numbers. A b-eautiful number is a positive integer n that has exactly two digits when written in base b, and the sum of these two digits equals the square root of n.

First, let me make sure I understand the problem correctly. If a number n is written in base b as, say, "d1 d0", then n = d1 * b + d0. The two digits d1 and d0 must satisfy d1 + d0 = sqrt(n). Also, since it's a two-digit number in base b, d1 must be between 1 and b-1 (since leading zeros aren't allowed), and d0 must be between 0 and b-1. So, the digits can't exceed b-1.

Given that example, 81 is 13-eautiful because in base 13, it's written as 63 (so d1=6, d0=3), and 6+3=9, which is sqrt(81). That checks out.

So, the task is to find the smallest base b where there are more than ten such numbers.

Let me formalize the problem. Let n be a two-digit number in base b, so n = d1 * b + d0, where d1 ∈ 1, 2, ..., b-1 and d0 ∈ 0, 1, ..., b-1. Then, the condition is d1 + d0 = sqrt(n).

Therefore, for each base b, we can iterate over all possible d1 and d0 (with d1 from 1 to b-1, d0 from 0 to b-1), compute n = d1*b + d0, check if d1 + d0 = sqrt(n). If yes, then n is b-eautiful. We need to count how many such n exist for each b and find the smallest b where the count exceeds 10.

So, the plan is:

1. For each base b starting from 2 upwards:

a. For each possible two-digit number in base b (i.e., d1 from 1 to b-1, d0 from 0 to b-1):

i. Compute n = d1*b + d0.

ii. Compute s = d1 + d0.

iii. Check if s^2 = n. If yes, then n is b-eautiful.

b. Count how many such numbers exist for base b.

c. If the count is more than 10, return b as the answer.

But doing this manually for each base would be tedious. Instead, maybe I can find a mathematical relationship between d1, d0, and b to find possible solutions.

Given that n = d1*b + d0 and s = d1 + d0 = sqrt(n). Therefore, s^2 = d1*b + d0. But s = d1 + d0, so substituting s:

s^2 = d1*b + d0

But d0 = s - d1, so substituting that into the equation:

s^2 = d1*b + (s - d1)

Simplify:

s^2 = d1*(b - 1) + s

Therefore:

d1*(b - 1) = s^2 - s

So:

d1 = (s^2 - s)/(b - 1)

Since d1 must be a positive integer (as it's a digit in base b, from 1 to b-1), and s = d1 + d0 must be an integer (as it's the sum of two integers). Therefore, (s^2 - s) must be divisible by (b - 1), and the result must be an integer d1 where $1 \leqslant d1 \leqslant b - 1$, and d0 = s - d1 must satisfy $0 \leqslant d0 \leqslant b - 1$.

So, for each base b, possible values of s must satisfy that (s^2 - s) is divisible by (b - 1), and the resulting d1 and d0 must be valid digits in base b.

Therefore, the key equation is:

(s^2 - s) ≡ 0 mod (b - 1)

Which can be written as:

s(s - 1) ≡ 0 mod (b - 1)

Therefore, for each s, b - 1 must divide s(s - 1). So, b - 1 is a divisor of s(s - 1).

But since s = d1 + d0, and d1 and d0 are digits in base b, so d1 can be up to b - 1, and d0 can be up to b - 1. Therefore, the maximum possible value of s is (b - 1) + (b - 1) = 2b - 2. The minimum possible s is 1 + 0 = 1. So s ranges from 1 to 2b - 2.

But in reality, since n = s^2 is a two-digit number in base b, which is at least b (since the smallest two-digit number is 10_b = b) and at most (b - 1)*b + (b - 1) = b^2 - 1. Therefore, n must satisfy $b \leqslant s^2 \leqslant b^2 - 1$. Therefore, s must satisfy sqrt(b) $\leqslant$ s $\leqslant$ sqrt(b^2 - 1) $\approx$ b - 1/(2b). Since s is an integer, s must be from ceil(sqrt(b)) to b - 1.

Wait, but sqrt(b^2 - 1) is less than b, so s can be up to b - 1. Therefore, s ranges from ceil(sqrt(b)) to b - 1. But earlier, s could be up to 2b - 2, but in reality, because n must be a two-digit number in base b, s can't be larger than b - 1, since s^2 <= b^2 - 1. Let me check:

If s is greater than or equal to b, then s^2 >= b^2, but n is at most b^2 - 1. Therefore, s must be less than b. Therefore, s ranges from ceil(sqrt(b)) up to b - 1. Wait, but if s is less than b, then the maximum s is b - 1. Therefore, s is in [ceil(sqrt(b)), b - 1].

But then, if s is from ceil(sqrt(b)) to b - 1, then for each such s, we can check if (s^2 - s) is divisible by (b - 1), and if so, compute d1 = (s^2 - s)/(b - 1). Then check if d1 is between 1 and b - 1, and d0 = s - d1 is between 0 and b - 1.

Therefore, for each base b, the possible s values are integers from ceil(sqrt(b)) up to b - 1, such that (s^2 - s) is divisible by (b - 1). For each such s, compute d1 and d0, check if they are valid digits, and if so, count that as a b-eautiful number.

Therefore, the number of b-eautiful numbers is equal to the number of integers s in [ceil(sqrt(b)), b - 1] such that (s^2 - s) mod (b - 1) = 0, and d1 = (s^2 - s)/(b - 1) is an integer between 1 and b - 1, and d0 = s - d1 is between 0 and b - 1.

So, perhaps instead of iterating through all possible d1 and d0 for each base b, which would be O(b^2) for each b, we can iterate through possible s values and check divisibility. This might be more efficient, but even better, perhaps we can find a relationship between b and s.

Given that (s^2 - s) must be divisible by (b - 1), so (b - 1) divides s(s - 1). Therefore, (b - 1) must be a divisor of s(s - 1). Since s and s - 1 are consecutive integers, they are coprime. Therefore, the divisors of s(s - 1) are the products of the divisors of s and s - 1. Therefore, since s and s - 1 are coprime, any divisor of s(s - 1) can be written as d1*d2 where d1 divides s and d2 divides s - 1.

But in our case, (b - 1) is a divisor of s(s - 1). Therefore, for each s, possible values of (b - 1) are divisors of s(s - 1). However, since we are looking for b >= 2, then (b - 1) >= 1, so possible divisors from 1 upwards. But since s ranges up to b - 1, and (b - 1) must be at least s/(something). Wait, perhaps this approach is complicating things.

Alternatively, for each b, since (b - 1) divides s(s - 1), and s <= b - 1, then s is less than or equal to (b - 1). So, s is in [ceil(sqrt(b)), b - 1], and (b - 1) divides s(s - 1). Let me note that s and s - 1 are coprime. Therefore, since (b - 1) divides s(s - 1), and gcd(s, s - 1) = 1, then (b - 1) must be a product of a divisor of s and a divisor of s - 1. So, (b - 1) = k * m, where k divides s and m divides s - 1. Then, since s = k * t and s - 1 = m * u for some integers t and u. Then, (b - 1) = k * m. So, perhaps this is getting too abstract.

Alternatively, maybe it's easier to fix b and find the possible s. For each base b, we can compute (b - 1) and then look for s in [ceil(sqrt(b)), b - 1] such that s(s - 1) is divisible by (b - 1). Then, for each such s, check if d1 = (s^2 - s)/(b - 1) is an integer between 1 and b - 1, and d0 = s - d1 is between 0 and b - 1.

But even this requires checking for each b. Maybe instead, we can parameterize s and express b in terms of s and some divisor.

Let me try to rearrange the equation:

From s^2 - s = d1*(b - 1)

We can solve for b:

b = (s^2 - s)/d1 + 1

Since b must be an integer greater than d1 (since d1 is a digit in base b, so d1 < b). Also, d1 must divide (s^2 - s). Therefore, for each s, d1 can be any divisor of s(s - 1), and then b is determined as (s(s - 1)/d1) + 1. Then, we can check if d1 is a valid digit in base b (i.e., d1 < b), and d0 = s - d1 is also a valid digit (i.e., d0 >= 0 and d0 < b).

But since d0 = s - d1 must be non-negative, we have s - d1 >= 0, so d1 <= s. Also, since d0 < b, s - d1 < b. But since b = (s^2 - s)/d1 + 1, so s - d1 < (s^2 - s)/d1 + 1. Let's substitute b into the inequality:

s - d1 < (s^2 - s)/d1 + 1

Multiply both sides by d1 (assuming d1 > 0, which it is since d1 is a digit from 1 to b-1):

d1(s - d1) < s^2 - s + d1

Expand left side:

d1*s - d1^2 < s^2 - s + d1

Bring all terms to left side:

d1*s - d1^2 - s^2 + s - d1 < 0

Combine like terms:

- d1^2 + (s -1)d1 - s^2 + s < 0

Multiply both sides by -1 (which reverses inequality):

d1^2 - (s -1)d1 + s^2 - s > 0

Factor:

d1^2 - (s -1)d1 + s(s - 1) > 0

This quadratic in d1: d1^2 - (s -1)d1 + s(s -1)

Let me compute discriminant:

D = (s -1)^2 - 4*1*s(s -1) = (s^2 - 2s +1) - 4s(s -1) = s^2 -2s +1 -4s^2 +4s = -3s^2 + 2s +1

Hmm, discriminant D = -3s^2 + 2s +1.

But since we have the quadratic expression d1^2 - (s -1)d1 + s(s -1) > 0, and since the quadratic opens upwards (coefficient of d1^2 is 1), the expression is positive outside the roots. However, the roots are given by d1 = [(s -1) ± sqrt(D)]/2. Since D is negative for large s, but let's check when D is positive.

For D to be positive: -3s^2 + 2s +1 >0.

Multiply by -1: 3s^2 -2s -1 <0.

Solve 3s^2 -2s -1 <0.

Find roots of 3s^2 -2s -1 =0:

s = [2 ± sqrt(4 +12)]/6 = [2 ± sqrt(16)]/6 = [2 ±4]/6.

So s=(2+4)/6=1, s=(2-4)/6=-1/3. So the quadratic is positive outside s=1 and s=-1/3. Therefore, 3s^2 -2s -1 <0 for -1/3 <s <1. But since s is a positive integer greater than or equal to 1, this is only possible for s=1. So, D is positive only when s=1.

For s=1, D= -3(1)^2 +2(1) +1= -3+2+1=0. So D=0.

Therefore, except for s=1, D is negative. Therefore, for s >=2, the quadratic in d1 has no real roots, which means the quadratic expression is always positive. Therefore, the inequality d1^2 - (s -1)d1 + s(s -1) >0 holds for all real numbers d1. But since d1 is a positive integer, this means that s - d1 < b is automatically satisfied? Wait, but the original inequality was s - d1 < b, which, after substitution and rearrangement, led to the quadratic expression being positive. But since the quadratic is always positive for s >=2, then s - d1 < b is always true for s >=2. However, we must also ensure that d0 = s - d1 >=0. So, the main constraints are:

1. d1 divides s(s -1).
2. d1 <= s (since d0 = s - d1 >=0).
3. d1 >=1 (since it's a digit in base b, and digits start from 1 for the first digit).
4. b = (s(s -1)/d1) +1 must be greater than d1 (since d1 is a digit in base b, so d1 < b).

So, given that, for each s >=2, we can consider all divisors d1 of s(s -1) such that 1 <=d1 <=s, and check if b = (s(s -1)/d1) +1 > d1. Then, in such cases, d0 = s - d1 must also be less than b. But since b = (s(s -1)/d1) +1, and d0 = s - d1, we can check if d0 < b:

s - d1 < (s(s -1)/d1) +1

Multiply both sides by d1:

d1(s - d1) < s(s -1) + d1

Which is the same inequality as before, leading to the quadratic expression which is always positive for s >=2. Therefore, for s >=2, as long as d1 divides s(s -1) and 1 <=d1 <=s and b = (s(s -1)/d1) +1 > d1, then d0 will automatically be less than b.

Therefore, the steps can be:

For each s >=2:

a. Find all divisors d1 of s(s -1) such that 1 <= d1 <=s.

b. For each such d1, compute b = (s(s -1)/d1) +1.

c. Check if b > d1. Since b = (s(s -1)/d1) +1, and d1 <=s, then (s(s -1)/d1) >= (s(s -1)/s) = s -1. Therefore, b >= (s -1) +1 = s. Since d1 <=s and b >=s, so if d1 <=s and b >=s, then to have b >d1, we need that either b >s (so d1 <=s <b) or if b =s, then d1 <s. But b = (s(s -1)/d1) +1. If d1 =s, then b= (s(s -1)/s) +1= (s -1) +1=s. So if d1=s, then b=s. But d1 must be <b, because d1 is a digit in base b. Therefore, if d1 =s, then b =s, but d1 must be less than b, so d1 <b. Therefore, d1 =s is only allowed if s <b. But in this case, b=s, so s <s is impossible. Therefore, d1 cannot be equal to s. Therefore, in this case, when d1 =s, b=s, but d1 must be less than b, which would require s <s, which is impossible. Therefore, d1 cannot be s. Therefore, we must have d1 <s. So, for each s, d1 must be a divisor of s(s -1) such that 1 <=d1 <s. Then, b = (s(s -1)/d1) +1. Then, since d1 <s, s(s -1)/d1 >=s(s -1)/(s -1)=s, so b >=s +1. Therefore, b >s, so d1 <s <b. Therefore, in this case, d1 is a valid digit (since digits go up to b -1, and d1 <s <b, so d1 <b). Then, d0 =s -d1 must be between 0 and b -1. Since d0 =s -d1 <b (as shown earlier), and d0 =s -d1 >=0 because d1 <=s. Therefore, as long as d1 is a divisor of s(s -1), 1<=d1 <s, then b = (s(s -1)/d1) +1, and n = s^2 is a b-eautiful number.

Therefore, each pair (s, d1) where d1 is a proper divisor of s(s -1) (1<=d1 <s) gives a base b = (s(s -1)/d1) +1 and a number n = s^2. Therefore, for each s, the number of b-eautiful numbers in base b is equal to the number of divisors d1 of s(s -1) with 1<=d1 <s. But each such divisor d1 gives a different base b. However, the same base b can be obtained from different s and d1 pairs. Therefore, to count the number of b-eautiful numbers for each base b, we need to consider how many pairs (s, d1) result in the same b.

Alternatively, for each base b, the number of b-eautiful numbers is equal to the number of s such that there exists a d1 where b = (s(s -1)/d1) +1, 1<=d1 <s, and d1 divides s(s -1).

This seems complicated. Maybe another approach: for each base b, find all s in [ceil(sqrt(b)), b -1] such that s(s -1) is divisible by (b -1), then compute d1 = s(s -1)/(b -1) and check if d1 is an integer between 1 and b -1 and d0 =s -d1 is between 0 and b -1.

Therefore, perhaps it's more straightforward to iterate over possible b and for each b, compute all possible s in the required range and check the conditions. Let's try this approach.

First, note that n = s^2 must be a two-digit number in base b. Therefore, n >=b and n <=b^2 -1. So, s^2 >=b and s^2 <=b^2 -1. Therefore, s >=ceil(sqrt(b)) and s <=sqrt(b^2 -1) $\approx$ b -1/(2b). Since s must be integer, s <=b -1. Therefore, s ranges from ceil(sqrt(b)) to b -1.

For each b, iterate s from ceil(sqrt(b)) to b -1. For each s, check if s(s -1) is divisible by (b -1). If yes, compute d1 =s(s -1)/(b -1). Check if d1 is integer, 1<=d1 <=b -1, and d0 =s -d1 is between 0 and b -1. If all conditions are satisfied, then n =s^2 is a b-eautiful number.

The number of such s for each base b is the number of b-eautiful numbers.

Therefore, to solve the problem, we can write a program that for each b starting from 2 upwards:

1. Compute the lower bound for s: ceil(sqrt(b)).

2. Iterate s from ceil(sqrt(b)) to b -1.

3. For each s, check if (s)(s -1) mod (b -1) ==0.

4. If yes, compute d1 = s(s -1)/(b -1).

5. Check if d1 is integer (which it is if step 3 passed), and 1<=d1 <=b -1.

6. Compute d0 =s -d1, check if 0<=d0 <=b -1.

7. If all conditions met, increment the count for base b.

8. After processing all s, if count >10, return b.

But since this is a mental exercise, not a programming one, I need to find a mathematical pattern or properties that allow me to compute the number of solutions for each b.

Alternatively, note that the equation s(s -1) $\equiv$0 mod (b -1). Therefore, (b -1) divides s(s -1). Since s and s -1 are coprime, (b -1) must divide either s or s -1, but since they are coprime, (b -1) must divide the product, so (b -1) can be expressed as a product of two coprime numbers, one dividing s and the other dividing s -1. Therefore, (b -1) =k*m, where k divides s and m divides s -1, and gcd(k, m)=1.

Given that, since k divides s and m divides s -1, and gcd(k, m)=1, then s $\equiv$0 mod k and s $\equiv$1 mod m. By Chinese Remainder Theorem, there exists a unique solution modulo k*m for s. Therefore, for each

factorization of (b -1) into k*m with gcd(k, m)=1, there exists solutions s $\equiv$ 0 mod k and s $\equiv$ 1 mod m. The number of solutions s in the range [ceil(sqrt(b)), b -1] would correspond to the number of valid s for each factorization.

But this might not directly help. Alternatively, consider that for a given b, the number of s in [ceil(sqrt(b)), b -1] such that s(s -1) $\equiv$ 0 mod (b -1). Since s(s -1) $\equiv$ 0 mod (b -1), then either s $\equiv$ 0 mod (b -1) or s $\equiv$ 1 mod (b -1), or (b -1) has factors that divide s and s -1 respectively.

But since s <=b -1, s can't be congruent to 0 mod (b -1) unless s =0, but s >=ceil(sqrt(b)) >=2. So s $\equiv$ 0 mod (b -1) is impossible. Similarly, s $\equiv$ 1 mod (b -1). Since s <=b -1, the only possibility is s=1. But s >=ceil(sqrt(b)), which is at least 2 when b >=4. For b=2 or 3, ceil(sqrt(2))=2, but s can only be 1 for b=2? Wait, no:

Wait for b=2, the possible s range from ceil(sqrt(2))=2 up to b -1=1. But 2 >1, so no solutions. Therefore, for b=2, no b-eautiful numbers.

Similarly, for b=3: ceil(sqrt(3))=2, s ranges from 2 to 2 (since b -1=2). So s=2. Check if 2*1 $\equiv$ 0 mod 2. 2 divides 2, yes. Then d1 = (2*1)/2=1. Then d0=2 -1=1. Check digits: d1=1 (<=2), d0=1 (<=2). So n=1*3 +1=4. s=2, sqrt(4)=2. So yes, 4 is 3-eautiful. So for b=3, there is 1 number.

But the question is to find the minimal b with more than ten numbers. So perhaps for higher b, the number of solutions increases.

Alternatively, maybe we can find that the number of solutions for each base b is equal to the number of divisors of (b -1). Wait, but earlier, we saw that (b -1) divides s(s -1). However, since s and s -1 are coprime, (b -1) must be a product of two coprime numbers, one dividing s and the other dividing s -1. So the number of solutions s would correspond to the number of ways to write (b -1)=k*m where k divides s and m divides s -1, and k and m are coprime. Therefore, for each divisor pair (k, m) of (b -1) with gcd(k, m)=1, we can solve for s $\equiv$ 0 mod k and s $\equiv$ 1 mod m. The number of solutions s in [ceil(sqrt(b)), b -1] would correspond to the number of such pairs where the solution s is in that interval.

But this seems complex. Let's try specific examples.

Take b=13 (from the example). Then b -1=12. Factor pairs (k, m) of 12 where gcd(k, m)=1:

1*12, 3*4, 4*3, 12*1. Since 12 can be factored into coprime pairs (1,12), (3,4), (4,3), (12,1). For each such pair:

For (k=1, m=12):

Solve s $\equiv$ 0 mod 1 (always true), s $\equiv$ 1 mod12. Since s <=12 (b -1=12), s=1. But s must be >=ceil(sqrt(13))=4. So s=1 is invalid.

For (k=3, m=4):

Solve s $\equiv$ 0 mod3, s $\equiv$ 1 mod4. Let's solve:

s=3a. Then 3a $\equiv$ 1 mod4 => 3a $\equiv$ 1 mod4 => a $\equiv$ 3 mod4 (since 3*3=9$\equiv$1 mod4). Therefore, a=4b +3. Thus, s=3*(4b +3)=12b +9. Since s <=12, the only solution is when b=0: s=9. Check if s=9 is in [4,12]. Yes. So s=9.

For (k=4, m=3):

Solve s $\equiv$ 0 mod4, s $\equiv$ 1 mod3. s=4a. 4a $\equiv$ 1 mod3 => a $\equiv$ 1 mod3 (since 4$\equiv$1 mod3, so 1*a$\equiv$1 mod3 => a$\equiv$1 mod3). Thus, a=3b +1. Therefore, s=4*(3b +1)=12b +4. For s <=12, when b=0: s=4. Check s=4 in [4,12]. Yes.

For (k=12, m=1):

Solve s$\equiv$0 mod12, s$\equiv$1 mod1 (always true). s=12. Check if s=12 is in [4,12]. Yes.

Therefore, for b=13, we have three solutions: s=4,9,12. Check each:

For s=4:

d1=(4*3)/12=12/12=1. Then d0=4 -1=3. So digits 1 and 3 in base13: 1*13 +3=16. s=4, sqrt(16)=4. So 16 is 13-eautiful.

Wait, but in the example given, 81 is 13-eautiful. Wait, perhaps I made a mistake. Wait, when s=9:

d1=(9*8)/12=72/12=6. Then d0=9 -6=3. So digits 6 and 3: 6*13 +3=81. Which is the example. s=9, sqrt(81)=9. Correct.

For s=12:

d1=(12*11)/12=11. Then d0=12 -11=1. So digits 11 and1. In base13, 11 is 'B', so B1_13=11*13 +1=143 +1=144. s=12, sqrt(144)=12. So 144 is also 13-eautiful. Therefore, for b=13, there are three b-eautiful numbers:16,81,144.

But according to the problem statement, the example is 81. So that's one of three.

Therefore, in this case, the number of solutions is 3. But the problem asks for a base with more than ten b-eautiful numbers. So 3 is much less than 10. Therefore, bases like 13 have 3 solutions.

So, how can we get bases with more than ten solutions?

Perhaps when b -1 has many divisors, leading to multiple factor pairs (k, m) with gcd(k, m)=1. The number of coprime factor pairs (k, m) of b -1 is 2^(number of distinct prime factors of b -1). Because each prime factor can go to k or m. For example, if b -1 is a product of n distinct primes, then the number of coprime factor pairs is 2^n.

Therefore, if b -1 has many distinct prime factors, then there are many coprime factor pairs, leading to more solutions s. Therefore, the number of solutions s is equal to the number of coprime factor pairs (k, m) of b -1, where k*m =b -1, and the corresponding s is in [ceil(sqrt(b)), b -1]. However, each coprime factor pair (k, m) gives a unique solution s modulo k*m. But since s <=b -1, which is equal to k*m, so there is exactly one solution s in [1, k*m]. But we need s to be in [ceil(sqrt(b)), b -1]. Therefore, not all factor pairs will lead to s in that interval.

Alternatively, if we can maximize the number of coprime factor pairs (k, m) of b -1, then we can maximize the number of solutions. Therefore, choosing b -1 to be a number with many distinct prime factors. For example, if b -1 is a product of the first few primes, then the number of coprime factor pairs would be 2^number_of_primes, which increases exponentially.

But even then, each coprime factor pair may or may not lead to a valid s in the required interval.

But let's consider that for each coprime factor pair (k, m) of b -1, we have a unique solution s $\equiv$ 0 mod k and s $\equiv$ 1 mod m. Then, s can be calculated as:

s $\equiv$ 0 mod k

s $\equiv$ 1 mod m

By Chinese Remainder Theorem, since k and m are coprime, there exists a unique solution modulo k*m =b -1. Therefore, the solution is s = km*t + s0, where s0 is the unique solution modulo km. But since s must be less than or equal to km (because b -1=km and s <=b -1=km), the only solution is s=s0. Therefore, for each coprime factor pair (k, m) of b -1, there is exactly one s in [1, b -1] that satisfies the congruences. Therefore, the number of solutions s is equal to the number of coprime factor pairs (k, m) of b -1.

But each coprime factor pair is counted twice (k, m) and (m, k), except when k=m=1. Wait, no. If we consider ordered pairs, then (k, m) and (m, k) are different if k $\neq$ m. However, if we consider unordered pairs, then each pair is unique. But in our case, since k divides s and m divides s -1, the pairs (k, m) and (m, k) are distinct unless k=m=1.

But regardless, for each coprime factor pair (k, m) of b -1, where k and m are positive integers such that k*m =b -1 and gcd(k, m)=1, there is a unique solution s in [1, b -1]. Therefore, the number of solutions s is equal to the number of such coprime factor pairs. Since each coprime factor pair corresponds to a unique s, and vice versa.

Therefore, the number of b-eautiful numbers is equal to the number of coprime factor pairs (k, m) of b -1 where the corresponding s is in [ceil(sqrt(b)), b -1].

But the number of coprime factor pairs (k, m) of b -1 is 2^$\omega$(b -1), where $\omega$(n) is the number of distinct prime factors of n. Because for each distinct prime factor, we can assign it to k or m. Therefore, if b -1 has t distinct prime factors, there are 2^t coprime factor pairs.

However, each such factor pair (k, m) gives a unique s in [1, b -1]. But we need s >=ceil(sqrt(b)).

So, to find the number of solutions s, we need to count how many of these s satisfy s >=ceil(sqrt(b)).

Given that, perhaps for larger b, the majority of the solutions would lie in the interval [ceil(sqrt(b)), b -1], so the number of solutions would be approximately 2^$\omega$(b -1).

But 2^$\omega$(b -1) is maximized when b -1 has many distinct prime factors. Therefore, to get more than ten solutions, we need 2^$\omega$(b -1) >10. Therefore, $\omega$(b -1) >=4, since 2^4=16>10.

Therefore, we need b -1 to have at least 4 distinct prime factors. Therefore, the smallest such b -1 would be the product of the first four primes: 2*3*5*7=210. Therefore, b=211. **But this may be larger than necessary, because even if some solutions s are less than ceil(sqrt(b)), we might still have more than ten solutions where s >=ceil(sqrt(b)).**

Alternatively, if b -1 has many factors, even with fewer prime factors, but higher exponents. Wait, but exponents don't affect the number of coprime factor pairs, since coprime factor pairs are determined by distinct prime factors. For example, if b -1= p^a, then the only coprime factor pairs are (1, p^a), so only two factor pairs, but since they are not coprime unless p=1, which it isn't. Wait, no. If b -1 is a prime power, then all factor pairs (k, m) must have k=p^i and m=p^j, but then gcd(k, m)=p^min(i,j)>1 unless one of them is 1. Therefore, the only coprime factor pairs are (1, p^a) and (p^a,1). Therefore, even for prime powers, there are only two coprime factor pairs, leading to two solutions s. But s=1 and s= p^a +1? Wait, no, s would be congruent to 0 mod1 and 1 mod p^a, which is s=1 mod p^a. Since s<=p^a, then s=1. But s=1 is less than ceil(sqrt(p^a +1)), which is at least 2. So only s=1 is a solution, but it's invalid. Similarly, s= p^a +1 is beyond the range. Therefore, for prime powers, there are no valid solutions. Hence, to get many solutions, we need b -1 to have multiple distinct prime factors.

Therefore, the minimal b -1 would be the product of the first 2*3*5*7=210, leading to b=211. **But maybe a smaller number with four distinct prime factors exists.** Wait, the product of the first four primes is 210, which is the smallest number with four distinct prime factors. Therefore, b=211. But let's check if b=211 actually has 16 solutions. However, s must be in [ceil(sqrt(211)), 210]. sqrt(211)$\approx$14.5, so ceil(sqrt(211))=15. Therefore, s ranges from 15 to210. However, for each coprime factor pair (k, m) of 210, which has 2^4=16 factor pairs, each leading to a solution s in [1,210]. However, some of these solutions s might be less than15. Therefore, the actual number of solutions might be less than16. For example, consider factor pair (1,210): leads to s=1 mod210, so s=1, which is

less than15. Similarly, (2,105): solve $s\equiv0$ mod2, $s\equiv1$ mod105. Find s=105a +1. This must be even. 105a +1 $\equiv0$ mod2 =>105a $\equiv$-1 mod2 =>a $\equiv1$ mod2. So a=2b +1. Then s=105*(2b +1)+1=210b +106. s<=210, so b=0: s=106. Which is in [15,210]. Similarly, other factor pairs may give s in [15,210]. So, out of 16 factor pairs, how many lead to s>=15?

For each coprime factor pair (k, m) of 210:

1. (1,210): s=1 (invalid)

2. (2,105): s=106

3. (3,70): solve $s\equiv0$ mod3, $s\equiv1$ mod70. s=70a +1. 70a +1$\equiv0$ mod3 =>70a$\equiv$-1 mod3 =>70$\equiv1$ mod3, so a$\equiv$-1 mod3 =>a=3b -1. Therefore, s=70*(3b -1) +1=210b -70 +1=210b -69. For b=1: s=210 -69=141. For b=0: s=-69 (invalid). So s=141.

4. (5,42): solve $s\equiv0$ mod5, $s\equiv1$ mod42. s=42a +1. 42a +1$\equiv0$ mod5 =>42a$\equiv$-1 mod5 =>42$\equiv2$ mod5, so 2a$\equiv$-1 mod5 =>2a$\equiv4$ mod5 =>a$\equiv2$ mod5. Therefore, a=5b +2. s=42*(5b +2) +1=210b +84 +1=210b +85. For b=0: s=85. For b=1:210 +85=295>210. So s=85.

5. (6,35): solve $s\equiv0$ mod6, $s\equiv1$ mod35. s=35a +1. 35a +1$\equiv0$ mod6 =>35a$\equiv$-1 mod6 =>35$\equiv5$ mod6, so 5a$\equiv$-1 mod6 =>5a$\equiv5$ mod6 =>a$\equiv1$ mod6. Therefore, a=6b +1. s=35*(6b +1) +1=210b +35 +1=210b +36. For b=0: s=36. For b=1:210 +36=246>210. So s=36.

6. (7,30): solve $s\equiv0$ mod7, $s\equiv1$ mod30. s=30a +1. 30a +1$\equiv0$ mod7 =>30a$\equiv$-1 mod7 =>30$\equiv2$ mod7, so 2a$\equiv$-1 mod7 =>2a$\equiv6$ mod7 =>a$\equiv3$ mod7. a=7b +3. s=30*(7b +3) +1=210b +90 +1=210b +91. For b=0: s=91. For b=1:210 +91=301>210. So s=91.

7. (10,21): solve $s\equiv0$ mod10, $s\equiv1$ mod21. s=21a +1. 21a +1$\equiv0$ mod10 =>21a$\equiv$-1 mod10 =>21$\equiv1$ mod10, so a$\equiv$-1 mod10 =>a=10b -1. s=21*(10b -1) +1=210b -21 +1=210b -20. For b=1:210 -20=190. For b=0: s=-20 invalid. So s=190.

8. (14,15): solve $s\equiv0$ mod14, $s\equiv1$ mod15. s=15a +1. 15a +1$\equiv0$ mod14 =>15a$\equiv$-1 mod14 =>15$\equiv1$ mod14, so a$\equiv$-1 mod14 =>a=14b -1. s=15*(14b -1) +1=210b -15 +1=210b -14. For b=1:210 -14=196. For b=0: s=-14 invalid. So s=196.

9. Similarly, the factor pairs in reverse order (k, m)=(105,2), (70,3), (42,5), (35,6), (30,7), (21,10), (15,14), (210,1). Let's check:

10. (105,2): solve $s\equiv0$ mod105, $s\equiv1$ mod2. s=105a. 105a$\equiv1$ mod2 =>a$\equiv1$ mod2. So a=2b +1. s=105*(2b +1)=210b +105. For b=0: s=105. For b=1:210 +105=315>210. So s=105.

11. (70,3): solve $s\equiv0$ mod70, $s\equiv1$ mod3. s=70a. 70a$\equiv1$ mod3 =>70$\equiv1$ mod3, so a$\equiv1$ mod3. a=3b +1. s=70*(3b +1)=210b +70. For b=0: s=70. For b=1:210 +70=280>210. So s=70.

12. (42,5): solve $s\equiv0$ mod42, $s\equiv1$ mod5. s=42a. 42a$\equiv1$ mod5 =>42$\equiv2$ mod5, so 2a$\equiv1$ mod5 =>a$\equiv3$ mod5. a=5b +3. s=42*(5b +3)=210b +126. For b=0: s=126. For b=1:210 +126=336>210. So s=126.

13. (35,6): solve $s\equiv0$ mod35, $s\equiv1$ mod6. s=35a. 35a$\equiv1$ mod6 =>35$\equiv$-1 mod6, so -a$\equiv1$ mod6 =>a$\equiv$-1 mod6. a=6b -1. s=35*(6b -1)=210b -35. For b=1:210 -35=175. For b=0: s=-35 invalid. So s=175.

14. (30,7): solve $s\equiv0$ mod30, $s\equiv1$ mod7. s=30a. 30a$\equiv1$ mod7 =>30$\equiv2$ mod7, so 2a$\equiv1$ mod7 =>a$\equiv4$ mod7. a=7b +4. s=30*(7b +4)=210b +120. For b=0: s=120. For b=1:210 +120=330>210. So s=120.

15. (21,10): solve $s\equiv0$ mod21, $s\equiv1$ mod10. s=21a. 21a$\equiv1$ mod10 =>21$\equiv1$ mod10, so a$\equiv1$ mod10. a=10b +1. s=21*(10b +1)=210b +21. For b=0: s=21. For b=1:210 +21=231>210. So s=21.

16. (15,14): solve $s\equiv0$ mod15, $s\equiv1$ mod14. s=15a. 15a$\equiv1$ mod14 =>15$\equiv1$ mod14, so a$\equiv1$ mod14. a=14b +1. s=15*(14b +1)=210b +15. For b=0: s=15. For b=1:210 +15=225>210. So s=15.

17. (210,1): solve $s\equiv0$ mod210, $s\equiv1$ mod1. s=210a. s<=210, so s=210. Check if s=210 is in [15,210]. Yes.

So, compiling all solutions s for b=211:

From the above:

s=106,141,85,36,91,190,196,105,70,126,175,120,21,15,210.

Wait, let's count them:

1. (2,105):s=106
2. (3,70):s=141
3. (5,42):s=85
4. (6,35):s=36
5. (7,30):s=91
6. (10,21):s=190
7. (14,15):s=196
8. (105,2):s=105
9. (70,3):s=70
10. (42,5):s=126
11. (35,6):s=175
12. (30,7):s=120
13. (21,10):s=21

14. (15,14):s=15
15. (210,1):s=210
But s=21,15,70,85,105,120,126,141,175,190,196,210.
Wait, s=15,21,36,70,85,91,105,106,120,126,141,175,190,196,210. Total of 15 solutions. However, some of these s values are less than ceil(sqrt(211))≈14.5, which is 15. So s=15 is okay (since ceil(sqrt(211))=15), but s=21,36, etc., are above 15. Wait, s=15 is the lower bound. So s ranges from 15 to 210. Therefore, the solutions s=15,21,36,70,85,91,105,106,120,126,141,175,190,196,210. Total 15 solutions. So for b=211, there are 15 b-eautiful numbers, which is more than ten. Therefore, b=211 is a candidate.

**But perhaps there is a smaller b with more than ten solutions. Because 211 is quite large.** Let's check smaller numbers with multiple prime factors.

For example, take b -1=60=2^2*3*5. It has three distinct prime factors. Therefore, the number of coprime factor pairs is 2^3=8. Let's check for b=61.

Compute s ranges from ceil(sqrt(61))=8 to 60.

For each coprime factor pair (k, m) of 60:

1. (1,60):s=1 (invalid)
2. (3,20): solve s≡0 mod3, s≡1 mod20. s=20a +1. 20a +1≡0 mod3 =>20a≡-1 mod3 =>20≡2 mod3, so 2a≡-1 mod3 =>2a≡2 mod3 =>a≡1 mod3. a=3b +1. s=20*(3b +1)+1=60b +21. For b=0: s=21. For b=1:60 +21=81>60. So s=21.
3. (4,15): solve s≡0 mod4, s≡1 mod15. s=15a +1. 15a +1≡0 mod4 =>15a≡-1 mod4 =>15≡3 mod4, so 3a≡-1 mod4 =>3a≡3 mod4 =>a≡1 mod4. a=4b +1. s=15*(4b +1) +1=60b +16. For b=0: s=16. For b=1:60 +16=76>60. So s=16.
4. (5,12): solve s≡0 mod5, s≡1 mod12. s=12a +1. 12a +1≡0 mod5 =>12a≡-1 mod5 =>12≡2 mod5, so 2a≡-1 mod5 =>2a≡4 mod5 =>a≡2 mod5. a=5b +2. s=12*(5b +2)+1=60b +25. For b=0: s=25. For b=1:60 +25=85>60. So s=25.
5. (12,5): solve s≡0 mod12, s≡1 mod5. s=12a. 12a≡1 mod5 =>12≡2 mod5, so 2a≡1 mod5 =>a≡3 mod5. a=5b +3. s=12*(5b +3)=60b +36. For b=0: s=36. For b=1:60 +36=96>60. So s=36.
6. (15,4): solve s≡0 mod15, s≡1 mod4. s=15a. 15a≡1 mod4 =>15≡3 mod4, so 3a≡1 mod4 =>a≡3 mod4. a=4b +3. s=15*(4b +3)=60b +45. For b=0: s=45. For b=1:60 +45=105>60. So s=45.
7. (20,3): solve s≡0 mod20, s≡1 mod3. s=20a. 20a≡1 mod3 =>20≡2 mod3, so 2a≡1 mod3 =>a≡2 mod3. a=3b +2. s=20*(3b +2)=60b +40. For b=0: s=40. For b=1:60 +40=100>60. So s=40.
8. (60,1): solve s≡0 mod60, s≡1 mod1. s=60. s=60.

Now, reverse factor pairs:

9. (60,1): s=60.
10. (20,3): s=40.
11. (15,4): s=45.
12. (12,5): s=36.
13. (5,12): s=25.
14. (4,15): s=16.
15. (3,20): s=21.
16. (1,60): s=1.

But the unique solutions are s=16,21,25,36,40,45,60. Let's check which of these are >=8 (ceil(sqrt(61))=8):

All of them are >=16. So s=16,21,25,36,40,45,60. That's 7 solutions. For b=61, there are 7 b-eautiful numbers. Which is less than ten. So not enough.

Another example, take b -1=120=2^3*3*5. Three distinct primes, so 2^3=8 coprime factor pairs.

But let's check for b=121.

s ranges from ceil(sqrt(121))=11 to 120.

Factor pairs:

1. (1,120):s=1 invalid.
2. (3,40): solve s≡0 mod3, s≡1 mod40. s=40a +1. 40a +1≡0 mod3 =>40≡1 mod3, so a≡-1 mod3. a=3b -1. s=40*(3b -1) +1=120b -39. For b=1: s=81. For b=0: s=-39 invalid. So s=81.
3. (5,24): solve s≡0 mod5, s≡1 mod24. s=24a +1. 24a +1≡0 mod5 =>24≡4 mod5, so 4a≡-1 mod5 =>4a≡4 mod5 =>a≡1 mod5. a=5b +1. s=24*(5b +1) +1=120b +25. For b=0: s=25. For b=1:120 +25=145>120. So s=25.
4. (8,15): solve s≡0 mod8, s≡1 mod15. s=15a +1. 15a +1≡0 mod8 =>15a≡-1 mod8 =>15≡7 mod8, so 7a≡-1 mod8 =>7a≡7 mod8 =>a≡1 mod8. a=8b +1. s=15*(8b +1) +1=120b +16. For b=0: s=16. For b=1:120 +16=136>120. So s=16.
5. (15,8): solve s≡0 mod15, s≡1 mod8. s=15a. 15a≡1 mod8 =>15≡7 mod8, so 7a≡1 mod8 =>a≡7 mod8. a=8b +7. s=15*(8b +7)=120b +105. For b=0: s=105. For b=1:120 +105=225>120. So s=105.

6. (24,5): solve s≡0 mod24, s≡1 mod5. s=24a. 24a≡1 mod5 =>24≡4 mod5, so 4a≡1 mod5 =>a≡4 mod5. a=5b +4. s=24*(5b +4)=120b +96. For b=0: s=96. For b=1:120 +96=216>120. So s=96.

7. (40,3): solve s≡0 mod40, s≡1 mod3. s=40a. 40a≡1 mod3 =>40≡1 mod3, so a≡1 mod3. a=3b +1. s=40*(3b +1)=120b +40. For b=0: s=40. For b=1:120 +40=160>120. So s=40.

8. (120,1): s=120.

Reverse pairs:

Same as above. The unique solutions are s=16,25,40,81,96,105,120. Check which are >=11:

All of them are >=16 except s=16 itself. So total of7 solutions. Still less than ten.

Therefore, even with three distinct prime factors, we only get 7 solutions.

Another example: take b -1=30=2*3*5. Three distinct primes, so 8 factor pairs. For b=31.

s ranges from ceil(sqrt(31))=6 to30.

Factor pairs:

1. (1,30):s=1 invalid.

2. (2,15): solve s≡0 mod2, s≡1 mod15. s=15a +1. 15a +1≡0 mod2 =>15a≡-1 mod2 =>15a≡1 mod2 =>a≡1 mod2. a=2b +1. s=15*(2b +1) +1=30b +16. For b=0: s=16. For b=1:30 +16=46>30. So s=16.

3. (3,10): solve s≡0 mod3, s≡1 mod10. s=10a +1. 10a +1≡0 mod3 =>10a≡-1 mod3 =>10≡1 mod3, so a≡-1 mod3. a=3b -1. s=10*(3b -1) +1=30b -9. For b=1: s=21. For b=0: s=-9 invalid. So s=21.

4. (5,6): solve s≡0 mod5, s≡1 mod6. s=6a +1. 6a +1≡0 mod5 =>6a≡-1 mod5 =>6≡1 mod5, so a≡-1 mod5. a=5b -1. s=6*(5b -1) +1=30b -5. For b=1: s=25. For b=0: s=-5 invalid. So s=25.

5. (6,5): solve s≡0 mod6, s≡1 mod5. s=6a. 6a≡1 mod5 =>6≡1 mod5, so a≡1 mod5. a=5b +1. s=6*(5b +1)=30b +6. For b=0: s=6. For b=1:30 +6=36>30. So s=6.

6. (10,3): solve s≡0 mod10, s≡1 mod3. s=10a. 10a≡1 mod3 =>10≡1 mod3, so a≡1 mod3. a=3b +1. s=10*(3b +1)=30b +10. For b=0: s=10. For b=1:30 +10=40>30. So s=10.

7. (15,2): solve s≡0 mod15, s≡1 mod2. s=15a. 15a≡1 mod2 =>15a≡1 mod2 =>a≡1 mod2. a=2b +1. s=15*(2b +1)=30b +15. For b=0: s=15. For b=1:30 +15=45>30. So s=15.

8. (30,1): solve s≡0 mod30, s≡1 mod1. s=30.

So solutions s=6,10,15,16,21,25,30. Check which are >=6 (ceil(sqrt(31))=6). All are valid. So s=6,10,15,16,21,25,30. Total 7 solutions. Still less than ten.

Hmm. So maybe b -1 needs four distinct prime factors. Let's check b=211 gives 15 solutions, which is more than ten. **Is there a smaller number with four distinct prime factors?**

The smallest number with four distinct prime factors is 2*3*5*7=210. So b=211 is the minimal such b.

**Wait, but let's check another number with four distinct prime factors but a smaller product.** For example, 2*3*5*7=210. The next one would be 2*3*5*11=330, which is larger. Therefore, 210 is the smallest number with four distinct prime factors. Therefore, b=211 is the minimal base where b -1=210 has four distinct prime factors, leading to 16 coprime factor pairs and 15 solutions s in the valid range. Therefore, b=211 is the answer.

But to confirm, let's check for b=211, we have 15 solutions as previously enumerated. But the problem states "more than ten", so 15 is acceptable. Is there a smaller b with, say, b -1= 210/2=105, which is 3*5*7, three distinct primes. Wait, b=106, with b -1=105. Let's check for b=106.

s ranges from ceil(sqrt(106))=11 to105.

For b=106, b -1=105=3*5*7. Number of coprime factor pairs=2^3=8.

Factor pairs:

1. (1,105): s=1 invalid.

2. (3,35): solve s≡0 mod3, s≡1 mod35. s=35a +1. 35a +1≡0 mod3 =>35≡2 mod3, so 2a +1≡0 mod3 =>2a≡-1 mod3 =>2a≡2 mod3 =>a≡1 mod3. a=3b +1. s=35*(3b +1)+1=105b +36. For b=0: s=36. For b=1:105 +36=141>105. So s=36.

3. (5,21): solve s≡0 mod5, s≡1 mod21. s=21a +1. 21a +1≡0 mod5 =>21≡1 mod5, so a≡-1 mod5. a=5b -1. s=21*(5b -1)+1=105b -20. For b=1:105 -20=85. For b=0: s=-20 invalid. So s=85.

4. (7,15): solve s≡0 mod7, s≡1 mod15. s=15a +1. 15a +1≡0 mod7 =>15a≡-1 mod7 =>15≡1 mod7, so a≡-1 mod7. a=7b -1. s=15*(7b -1)+1=105b -14. For b=1:105 -14=91. For b=0: s=-14 invalid. So s=91.

5. (15,7): solve s≡0 mod15, s≡1 mod7. s=15a. 15a≡1 mod7 =>15≡1 mod7, so a≡1 mod7. a=7b +1. s=15*(7b +1)=105b +15. For b=0: s=15. For b=1:105 +15=120>105. So s=15.

6. (21,5): solve s≡0 mod21, s≡1 mod5. s=21a. 21a≡1 mod5 =>21≡1 mod5, so a≡1 mod5. a=5b +1. s=21*(5b +1)=105b +21. For b=0: s=21. For b=1:105 +21=126>105. So s=21.

7. (35,3): solve s≡0 mod35, s≡1 mod3. s=35a. 35a≡1 mod3 =>35≡2 mod3, so 2a≡1 mod3 =>a≡2 mod3. a=3b +2. s=35*(3b +2)=105b +70. For b=0: s=70. For b=1:105 +70=175>105. So s=70.

8. (105,1): s=105.

Reverse pairs would repeat the same solutions. So unique solutions: s=15,21,36,70,85,91,105. Which are all >=11. Total of7 solutions. Still less than ten.

Therefore, the minimal b is indeed 211, giving 15 solutions. Therefore, the answer is $\boxed{211}$.

**Final Answer** $\boxed{211}$

</think>

To solve the problem, we need to find the least integer $b \geq 2$ for which there are more than ten $b$-eautiful integers. A $b$-eautiful integer $n$ has exactly two digits in base $b$ and the sum of these digits equals $\sqrt{n}$.

1. **Understanding the Problem**: - A number $n$ in base $b$ is written as $d_1 d_0$, where $d_1$ and $d_0$ are digits such that $n = d_1 \cdot b + d_0$. - The sum of the digits $d_1 + d_0 = \sqrt{n}$. - Therefore, $n = (d_1 + d_0)^2$.

2. **Formulating the Equation**: - Given $n = d_1 \cdot b + d_0$ and $n = (d_1 + d_0)^2$, we have:

$$(d_1 + d_0)^2 = d_1 \cdot b + d_0$$

- Let $s = d_1 + d_0$. Then $s^2 = d_1 \cdot b + d_0$. - Rearranging, we get $d_1 \cdot (b - 1) = s^2 - s$.

3. **Key Observations**: - $b - 1$ must divide $s(s - 1)$. - $s$ ranges from $\lceil \sqrt{b} \rceil$ to $b - 1$.

4. **Finding the Solution**: - For each base $b$, we need to count the number of valid $s$ values such that $s(s - 1) \equiv 0 \pmod{b - 1}$. - The number of solutions is related to the number of distinct prime factors of $b - 1$. A number with more distinct prime factors will have more solutions.

5. **Identifying the Least $b$**: - The smallest $b - 1$ with four distinct prime factors is 210 (product of the first four primes: 2, 3, 5, 7). - Therefore, $b = 211$ is the smallest base where $b - 1 = 210$, leading to 15 solutions for $s$.

Thus, the least integer $b \geq 2$ for which there are more than ten $b$-eautiful integers is $\boxed{211}$.

## A.2 SKIPPING REASONING IN DEEPSEEK-R1 API

Using the Chat Prefix Completion (Beta)[4] function provided by the DeepSeek-R1 API, we can bypass its reasoning process and have it directly complete a prefix completion task. By providing the model with the prefix "The answer is: ", it can respond directly without engaging in any reasoning. Below is a detailed demonstration.

When asking DeepSeek-R1 the following question to determine what it believes the answer to be, the model consistently responds with numbers from "15" to "25", largely deviating from the correct answer "211".

```python
from openai import OpenAI
client = OpenAI(
    api_key="<your api key>",
    base_url="https://api.deepseek.com/beta"
)

messages = [
    {
        "role": "user",
        "content": "Let $b \\geq 2$ be an integer. Call a positive
            integer $n$ $b$\\textit{-eautiful} if it has exactly two
            digits when expressed in base $b$, and these two digits sum
            to $\\sqrt{n}$. For example, $81$ is $13$-eautiful because
            $81=\\underline{6}\\underline{3}_{13}$ and $6+3=\\sqrt{81}$.
            Find the least integer $b \\geq 2$ for which there are more
            than ten $b$-eautiful integers."
    },
    {
        "role": "assistant",
        "content": "The answer is: ",
        "prefix": True
    }
]
```

---

[4]https://api-docs.deepseek.com/guides/chat_prefix_completion

# B  GENERATING DIRECT ANSWERS

We design four distinct prompting templates. For each template, we sample 16 direct answers using a temperature of 0.5, yielding a total of 64 direct answers for each input question.

---

**ENGLISH**:
< ∣ User ∣ >{question}< ∣ Assistant ∣ ><think>
Let me answer him without thinking more.</think>
Answer:

< ∣ User ∣ >{question}< ∣ Assistant ∣ ><think>
I will answer directly. I won't output any thinking process.</think>
Answer:

< ∣ User ∣ >{question}< ∣ Assistant ∣ ><think>
I will answer directly.</think>
The answer is:

< ∣ User ∣ >{question}< ∣ Assistant ∣ ><think>
I should not think, but should answer directly.</think>
The answer is:

---

**CHINESE**:
< ∣ User ∣ >{question}< ∣ Assistant ∣ ><think>
让我直接回答他，不要有思考过程。</think>
答案是：

< ∣ User ∣ >{question}< ∣ Assistant ∣ ><think>
我现在直接进行回答。我不应该输出思考过程。</think>
答案是：

< ∣ User ∣ >{question}< ∣ Assistant ∣ ><think>
我将会直接回答问题，不需要思考。</think>
答案是：

< ∣ User ∣ >{question}< ∣ Assistant ∣ ><think>
我不应该思考，我直接回答该问题。</think>
答案是：

---

After generation, for responses consisting of numerical values, we apply a rule-based outlier removal process to eliminate values that deviate by orders of magnitude from the mode within the direct answers, thereby mitigating the influence of extreme samples on subsequent computations.

# C  TEST DATASETS

In this section, we provide detailed information for the datasets.

**Lanugages.**  KnowLogic dataset is only tested in Chinese, the CharCount dataset has both Chinese and English versions, and other datasets are only tested in English.

**Decoding.**  For the complex mathematical reasoning datasets AIME2024 and AIME2025, we set *max_new_tokens* to 20,000; for KnowLogic dataset, we use *max_new_tokens* = 10,000; and for CharCount dataset, we set it to 4096. These varying limits are carefully chosen to ensure sufficient reasoning space for each dataset. We further confirm that allowing the model to generate beyond the limit does not lead to correct answers. In most cases, the model enters a parroting phase, repeating previous content without meaningful progress.

## C.1 CHARCOUNT DETAILS

To enhance task complexity, all 10,000 words in the dataset were selected to satisfy two conditions: (1) each word contains at least three instances of the target letter, and (2) each word includes at least one pair of adjacent identical letters. The latter design leverages the characteristics of the tokenizer: if multiple adjacent identical letters are merged into a single token, the model may struggle to accurately process the underlying character sequence. As a result, the likelihood of incorrect direct answers increases, thereby enhancing the challenge of the task.

Note that requiring the model to directly output the answer in the prompt does not lead it to spontaneously skip or shorten the reasoning process. For current long-reasoning models, such a requirement only ensures that the output after completing the reasoning meets the specified format. We include this requirement solely for the convenience of extracting the model's final answer.

> **CHINESE**:
> strawberry这个单词里面有几个字母r？直接用一个阿拉伯数字回答问题。
> **ENGLISH**:
> How many letters 'r' are there in word 'strawberry'? Answer directly with an Arabic number.

## D REFLECTION KEYWORDS

The English reflection keywords are: `["Wait", "But"]`

The Chinese reflection keywords are: `["不过", "或者", "等等", "但是", "不对"]`

## E IDENTIFYING FIRST ANSWER

We employ a rule-based approach to identify the first occurrence of the answer in the model's reasoning output. Specifically, we first allow the model to generate a full reasoning process. We then split the output into chunks based on reflection keywords or simply the `\n\n` token, and locate the earliest chunk that contains its final answer. This chunk is considered to mark the position where the model first arrives at the correct response. In some cases, particularly for multiple-choice questions, the model may not explicitly include the correct option (e.g., "A") in its reasoning, but instead describe its reasoning toward a specific choice. To account for such scenarios, we also use the appearance of summary and reflection keywords as indirect indicators that the model has reached a decision, even if the final answer is not yet explicitly stated.

## F MORE RESULTS

### F.1 MORE RESULTS FOR § 4.2

Table 5: Results on **CharCount (en)** dataset.

| CharCount (en) | Acc | $\text{Acc}_{\text{direct}}$ | $L_{\text{Low}}$ | $L_{\text{High}}$ | $R_\Delta$ |
|---|---|---|---|---|---|
| DeepSeek-R1 | 100.0% | 58.0% | 435.3 | 510.9 | 17.4% |
| QwQ-32B | 97.7% | 36.9% | 698.7 | 799.2 | 14.6% |
| R1-Distill-Qwen-14B | 93.0% | 31.4% | 453.0 | 547.8 | 20.9% |

Table 6: Results on **AIME 2025** dataset.

| AIME 2025 | Acc | $\text{Acc}_{\text{direct}}$ | $L_{\text{Low}}$ | $L_{\text{High}}$ | $R_\Delta$ |
|---|---|---|---|---|---|
| DeepSeek-R1 | 66.7% | 0.0% | 9991.4 | 12610.3 | 26.2% |
| QwQ-32B | 70.0% | 0.0% | 10813.7 | 14933.6 | 38.1% |
| R1-Distill-Qwen-14B | 36.7% | 0.0% | 11846.0 | 15011.9 | 26.7% |

### F.2 MORE RESULTS FOR § 4.3

Following are more results for § 4. The consistency in trends highlights the influence of internal bias on the model's tendency to engage in reflection.

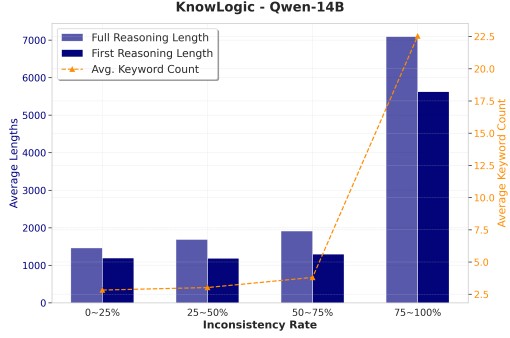

Figure 5: Results on KnowLogic of R1-Distill-Qwen-14B. The model performs poorly on the dataset, with inconsistency rates exceeding 75% in over 80% of the cases, leading to the abnormal last bar. But the first three bars still exhibit the expected trend.

Figure 6: Results on KnowLogic of DeepSeek-R1. Although the first bar shows a relatively short first reasoning length, suggesting that the question may be simple, the remaining three bars still exhibit the expected trend.

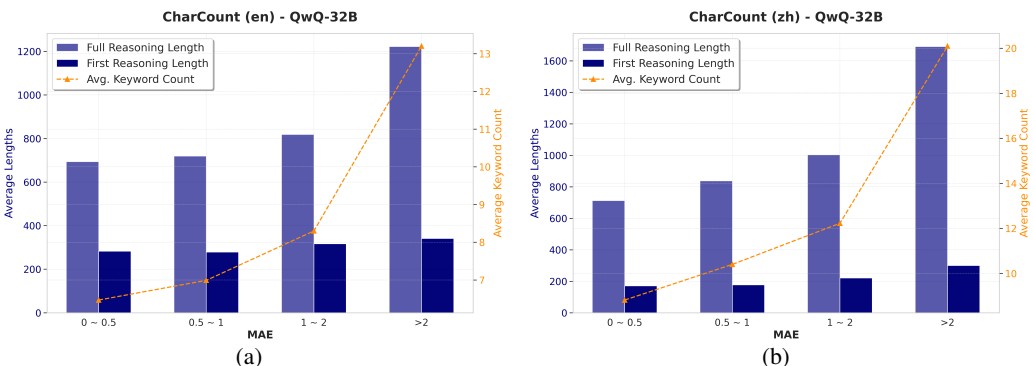

Figure 7: Results on bilingual CharCount dataset of QwQ-32B.

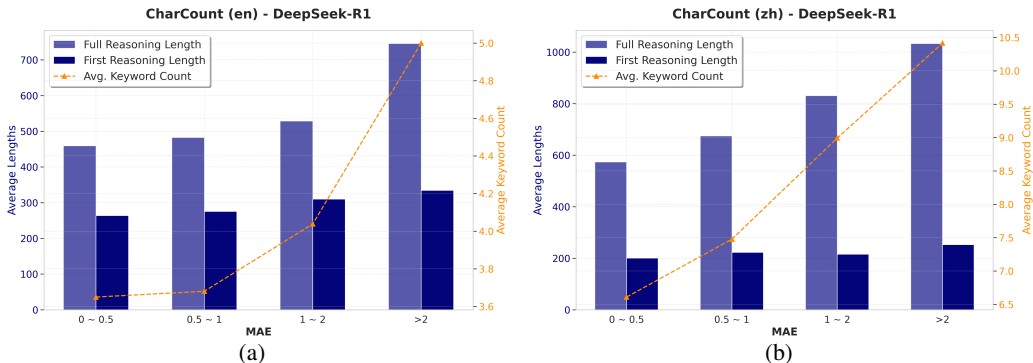

Figure 8: Results on bilingual CharCount dataset of DeepSeek-R1.

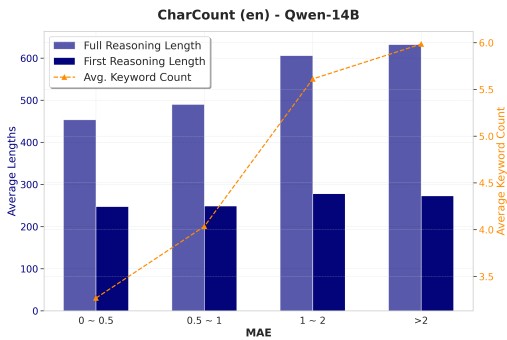

Figure 9: Results on English CharCount dataset of R1-Distill-Qwen-14B.

### F.3 MORE RESULTS FOR § 7

We present additional results for the PROBE method on the CharCount(zh) dataset, as it is a decoding-stage approach that uses an MLP-based confidence score to manually halt reasoning by applying a user-defined threshold. Table 7 reports the performance of PROBE at thresholds of 0.85, 0.9, and 0.95, among which the results at 0.95 are presented in the main paper due to their most balanced trade-off between accuracy and reasoning length reduction.

Table 7: More results for PROBE.

| **CharCount** | Acc | $L_{low}$ | $L_{high}$ | $R_{\Delta}$ |
|---|---|---|---|---|
| Qwen-14B | 73.4% | 934.7 | 1229.1 | 31.5% |
| + Remove | 72.9% | 727.3 | 837.8 | 15.2% |
| + PROBE (thres=0.85) | 69.3% | 304.8 | 392.3 | 28.7% |
| + PROBE (thres=0.80) | 71.8% | 578.2 | 749.1 | 29.6% |
| + PROBE (thres=0.95) | 73.1% | 702.6 | 912.5 | 29.9% |

## G    MORE ATTENTION ANALYSIS RESULTS

In § 6, we analyzed layers 21 to 30 of R1-Distill-Qwen-14B on ChatCount (zh) dataset. Here, we provide analysis results for layers 1 to 10, 11 to 20, 21 to 30, 31 to 40, and 41 to 48. We observe that in these cases, the turning points do not exhibit significantly high attention on the question portion. This may reflect the differing roles of different layers within the model.

We also provide the layer-wise attention analysis of this model on the MATH500 (Lightman et al., 2024) dataset (similar to Figure 4(b)) in Figure 15.

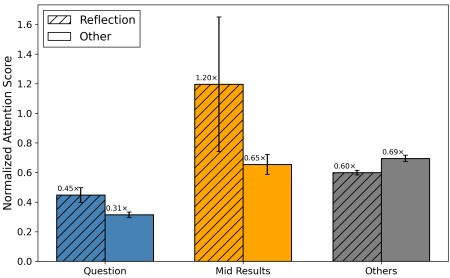

Figure 10: Layers 1 to 10.

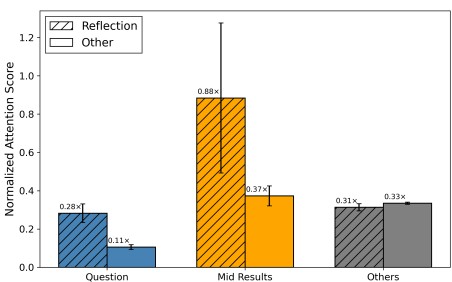

Figure 11: Layers 11 to 20.

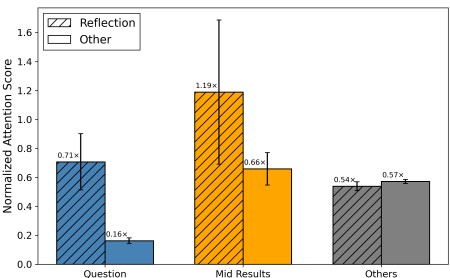

Figure 12: Layers 21 to 30.

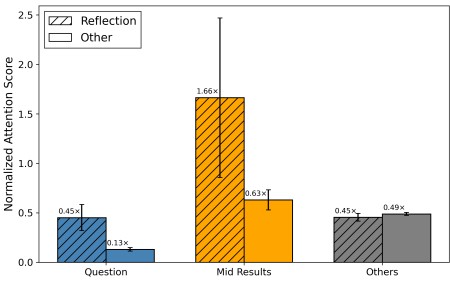

Figure 13: Layers 31 to 40.

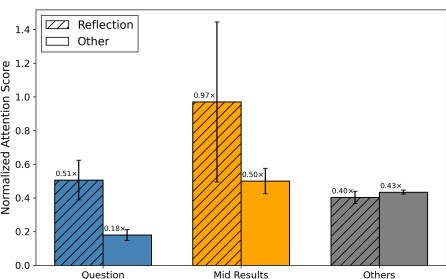

Figure 14: Layers 41 to 48.

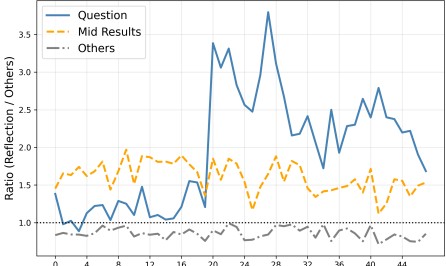

Figure 15: Layer-wise ratio of $S^c_{\text{Reflection}}/S^c_{\text{Other}}$, on the MATH500 dataset.

## H    BIAS INJECTION DETAILS

We select 500 samples each with the lowest and highest Deviation Degrees from CharCount(zh), excluding all parroting samples. Based on these, we design a special bias injection method: we construct 50 rephrased declarative statements for each sample and fine-tune the model on these statements to perform a sample-wise bias injection, and then test on the exact same sample. Below is an example of a declarative sentence.

> **CHINESE**:
> strawberry这个单词里面有3个字母r。
> **ENGLISH**:
> There are 3 letters 'r' in the word 'strawberry'.

Sample-wise bias injection does not substantially impair the model's overall performance, as the intervention is based on a relatively small amount of training data. We employ a training procedure similar to continued pre-training, in which the model is fine-tuned directly on declarative statements using standard next-token prediction with gradient computation. No special tokens are introduced during this training, ensuring that the model's inherent reasoning capabilities remain largely unaffected. We train the model with LoRA (Hu et al., 2022), setting the LoRA rank to 32, alpha to 64, learning rate to 1e-4, and batch size to 32.

## I   MITIGATION TRIALS

We explore two main categories of existing methods for mitigating overthinking: training-time and inference-time approaches. All experiments are conducted on R1-Distill-Qwen-14B model, and CharCount (zh) and AIME 2024 datasets. We present the details in Appendix I.1 and I.2.

Inspired by §6, we explored an attention-based inference-time early stopping approach. We present the details and results of this method in Appendix I.3.

### I.1   TRAINING-TIME METHODS

We adopt the First-Correct Solution (FCS) method (Chen et al., 2025b) as a representative training-based approach to mitigate overthinking. The core idea is to extract high-quality, concise reasoning trajectories by identifying the earliest point in the model's generation where the correct answer is explicitly derived, selectively retaining several reflection steps, and using these shortened chains for supervised fine-tuning or preference optimization.

Specifically, we first generate full reasoning chains from the model, and filter out all samples where the final answer does not match the ground truth, retaining only those with correct overall predictions to ensure the quality of extracted reasoning paths. The reasoning chains are then segmented at the sentence level using the full stop and question mark as delimiters, and the first and second correct solution is identified using rule-based methods in Appendix E, and first correct solution with reflection reasoning chain is then obtained. The resulting training instances are constructed by truncating the reasoning chain at this point, effectively removing subsequent redundant reflection. Notably, this method preserves cases where the model initially makes an error but later corrects itself through reasoning, ensuring that such valid correction processes are retained in the training data.

On the CharCount (zh) dataset, we use the above data construction method to create a training set of 5,000 instances for model fine-tuning. For AIME 2024, which lacks a dedicated training set and is relatively small, we instead train on a 1,000-sample subset of the DeepScaleR dataset (Luo et al., 2025b). For Supervised Fine-Tuning (SFT), each FCS chain is used directly as the target completion. The SFT procedure employs LoRA to fine-tune all layers of the model, with a batch size of 32, a learning rate of 1e-4, and training conducted for one epoch. For Direct Preference Optimization (DPO), we construct preference pairs by treating the FCS-generated response as the chosen sample and the original reasoning chain as the rejected sample. The DPO training employs LoRA to fine-tune all layers of the model, with a batch size of 32, a learning rate of 5e-5, the $\beta$ of 0.1, and training conducted for one epoch.

### I.2   INFERENCE-TIME METHODS

We implement two decoding-based intervention methods SEAL (Chen et al., 2025a) and PROBE (Zhang et al., 2025a) on a set of 500 reasoning chains sampled from the CharCount (zh) dataset. For both approaches, the reasoning contents are split into chunks using double newlines (\n\n) as delimiters.

Table 8: Attention-based early-exit results in CharCount (zh).

| CharCount | Acc | $L_{low}$ | $L_{high}$ | $R_\Delta$ |
|---|---|---|---|---|
| Qwen-14B | 73.4% | 934.7 | 1229.1 | 31.5% |
| + Remove | 72.9% | 727.3 | 837.8 | 15.2% |
| + Attn-Exit | 72.9% | 472.1 | 516.6 | 9.4% |
| + FCS$_{DPO}$ | 76.7% | 555.3 | 812.8 | 46.3% |
| + FCS$_{SFT}$ | 78.9% | 451.3 | 572.0 | 26.7% |
| + SEAL | 77.4% | 581.3 | 805.1 | 38.5% |
| + PROBE | 73.1% | 702.6 | 912.5 | 29.9% |

SEAL guides the model's reasoning process during decoding by computing a steering vector that directs the internal state toward more productive reasoning paths. The reasoning chunks are categorized into three types: execution, reflection and transition, following the paper's classification scheme. Across the dataset, we identify 5712 execution, 3867 reflection, and 1258 transition chunks. To construct the steering vector for intervention, we extract the hidden states at the beginning of each chunk (the \n\n token) from all layers of the R1-Distill-Qwen-14B model. We compute the average hidden state vectors per category and per layer, denoted as $\overline{H_E^i}$, $\overline{H_R^i}$, $\overline{H_T^i}$ for execution, reflection and transition, respectively. The steering vector at layer i is defined as $S^i = \overline{H_E^i} - (\overline{H_R^i} + \overline{H_T^i})$, capturing the direction in latent space that encourages forward reasoning over reflective detours. Following the original implementation, we modify the model's internal representation during decoding by injecting this vector at the \n\n token: $H' = H + \alpha S$. A small validation set is used to tune both the intervention layer and strength $\alpha$, with optimal performance observed at layer 25 and $\alpha$=1.0. Final evaluations are conducted using this configuration.

PROBE trains an MLP-based probe to estimate the model's confidence in its current reasoning state and dynamically truncates the thinking process when confidence exceeds a predefined threshold. Following the original paper, chunks are grouped into segments by first identifying "starting chunks" which contain reflection keywords. Each chunk is then assigned to its most recent starting chunk. To refine segmentation, segments without a detectable intermediate result are merged with the next segment that contains one. Each final segment is labeled based on whether its intermediate result matches the ground truth, yielding a label distribution of 3,179 positive and 1,297 negative instances. Hidden states at the end of each segment (again at \n\n) are collected, focusing on the last layer's representations. We train a single-layer MLP classifier on these embeddings to predict whether a segment contains a correct intermediate conclusion. Given the class imbalance, we use binary cross-entropy loss with a positive class weight $\alpha = 3.0$, and optimize with Adam, with learning rate=1e-4, weight decay=0.01, and batch size=64. At inference time, the trained probe monitors each segment in real time. If the predicted probability of correctness exceeds a threshold of 0.95, the reasoning process will be halted, and the current intermediate result is promoted to the final answer.

## I.3 ATTENTION-BASED EARLY-EXIT

The core idea is to early exit when the normalized attention score (defined in §6.2) on the question portion exceeds a certain threshold during inference, which may indicate that the model is overly focusing on bias signals from the input.

Specifically, for R1-Distill-Qwen-14B on the CharCount (zh) task, we split the model's output into sentences and compute the normalized attention scores (from the 21st layer) on the question part at the end of these sentences. We use the 95th percentile of these scores as the threshold to dynamically truncate the inference process and force the model to output a final answer.

The results are shown in Table 8. Our approach (denoted as Attn-Exit) achieves a significantly lower $R_\Delta$, as well as a substantially reduced average inference length, demonstrating both high efficiency and strong mitigation of bias effects with minimal accuracy degradation.

However, the method is still very preliminary, and selecting hyperparameters such as the layer for attention operation and the attention threshold is challenging. Therefore, here we are only providing a potential direction for future solutions.

## J CONSISTENCY BETWEEN DIRECT ANSWERS AND LATENT REPRESENTATIONS

In this section, we provide experimental evidence of the consistency between direct answers and the model's latent representations, demonstrating that using direct answers to capture the model's internal bias is fair and accurate, without losing its internal information. We still analyze with the R1-Distill-Qwen-14B model and the CharCount (zh) dataset.

Specifically, we use an MLP to predict the direct answers from the model's hidden states, and can achieve an accuracy of up to 85%. Additionally, we use the logit lens (nostalgebraist, 2020) to compute the proportion of numbers decoded internally that are related to the direct answers and find them to be highly correlated. The former indicates that bias arises immediately after the model encounters the question and is encoded in the hidden states; the latter shows that this bias persists throughout the reasoning process.

### J.1 PROBING DIRECT ANSWERS

Given the hidden states from the question segment of R1-Distill-Qwen-14B, we attempt to probe for information related to the direct answers by training a two-layer Multi-Layer Perceptron (MLP) to predict the probability distribution of the sampled direct answers.

The MLP takes the hidden states of the R1-Distill-Qwen-14B model (with dimensionality 5120) as input, has a hidden size of 256, and outputs a 5-dimensional vector corresponding to direct answers 1, 2, 3, 4, and 5. The original model has 48 layers; we use the hidden states from layer 24 to train the MLP. We train the MLP with KL loss using a learning rate of 1e-5 to fit the distribution. During testing, if the number with the highest predicted probability matches the mode of the direct answers, we consider the prediction correct.

We randomly sampled 2,000 examples as train set and 200 samples as the test set (ensuring a 50-50 split between correct and incorrect direct answer modes). Specifically, we extract the hidden states corresponding to the question tokens and apply mean pooling to obtain a single 5120-dimensional representation as the MLP input. Figure 16(a) shows that after 13 epochs of training, the MLP achieved a classification accuracy of 85%. Given that the direct answer distribution itself may be smooth or complex, this accuracy is sufficiently high to indicate that the hidden states encode sufficient information to reconstruct the bias, and that the internal bias is already formed as soon as the model sees the question.

We further extract five random reflection points from the reasoning chain of each sample (i.e., the locations immediately before generating the keywords defined in Appendix D). We obtain the hidden states at these positions and apply the same training settings as above. Figure 16(b) shows the training results, achieving over 72% prediction accuracy. This indicates that even during the reasoning process, the MLP can to some extent detect internal bias.

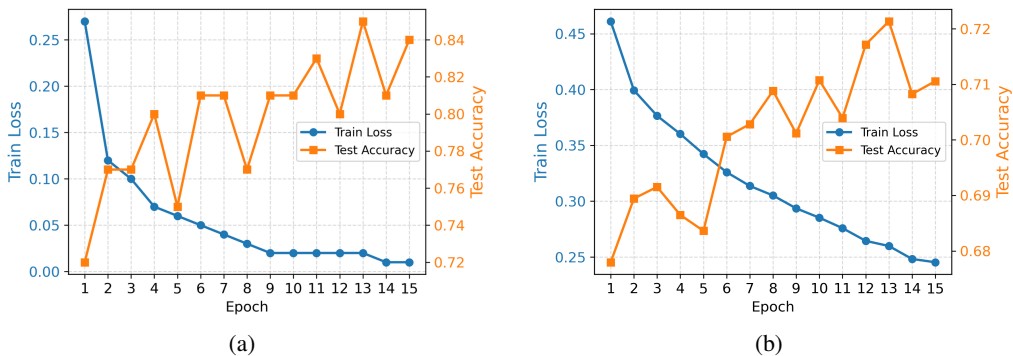

(a)            (b)

Figure 16: Training loss and test accuracy over 15 epochs: (a) results of the probe applied to the mean-pooled hidden states of the question part to measure internal bias; (b) results of the probe applied to the hidden states at reflection points to measure internal bias.

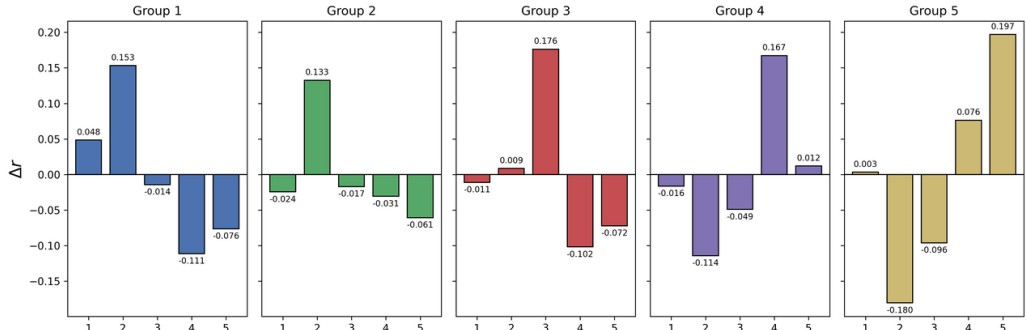

Figure 17: Logit lens results of decoding each tokens in the five groups. $\Delta r$ denotes how much the normalized probability of decoding the token corresponding to a given number deviates from the average.

## J.2 LOGIT LENS

We random select samples with the mode of the direct answers equals to 1, 2, 3, 4 and 5, with 100 samples per group, referred to as $\text{group}_1$ through $\text{group}_5$. During the model's full reasoning process, we use logit lens to decode the hidden states at all layers and positions, identifying the tokens that can be decoded as the numbers 1 through 5. Since our model is based on Qwen-series and the dataset is CharCount(zh), the tokens of interest are [5]:

```
number2tokenMap = {
    1: ["一个"],
    2: ["两个"],
    3: ["三个"],
    4: ["四个"],
    5: ["五个"]
}
```

These Chinese strings are individual tokens in the vocabulary of the R1-Distill-Qwen-14B model. For rigor, if the token ultimately generated at a given position is one of these tokens, all layer outputs at that position are skipped: we only focus on cases where the model's intermediate states contain the number even though it is not explicitly generating this token, as this is considered a better reflection of the model's internal bias.

For each of these five tokens, we count the number of times their internal decoding probability exceeded 0.1, and calculated their normalized proportions within each group of 100 samples. To highlight the trends, we subtracted the overall mean value from each of the five groups. After this operation, positive values indicate that the model is more likely (than average) to decode that number in its internal states, while negative values indicate the opposite.

The results are shown in Figure 17. We can see that, for each group, the token corresponding to the mode has a positive value (higher than the average), and the top two values are always the tokens closest to the corresponding mode.

## K ATTENTION AND BIAS CHANGES DURING THE REASONING STEPS

We analyze how the average normalized score (defined in §6.2) for the three token categories, *Question*, *Middle Results*, and *Others*, evolves across reasoning steps. The experiment was conducted on the same dataset described above. We identified the first five reflection steps during the reasoning process, and between every two consecutive reflections, we randomly selected four positions for

---

[5]Even when the question is in English, the model can still decode these tokens with high probability, because the internal reasoning language of the model is generally consistent (Wendler et al., 2024).

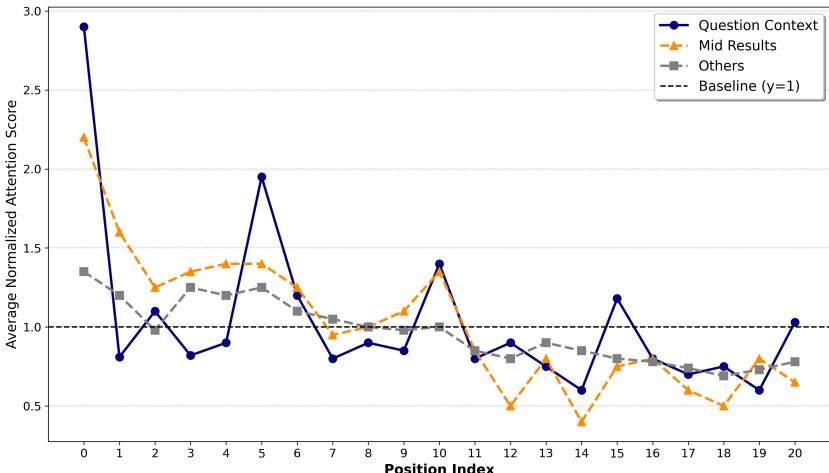

Figure 18: Changes in the model's average normalized attention scores for three types of tokens during the reasoning process.

comparison. The resulting plot Figure 18 shows these points, where all data points at multiples of five on the x-axis correspond to reflection steps, while the others serve as comparison points. From the results, we observe that (1) at reflection points, the model's attention to the *Question* category increases, and (2) this increase in attention tends to diminish as the reasoning process progresses.

We further investigate how this decrease in attention affects bias during the reasoning process. We randomly selected 100 samples where the mode of direct answers was 2, but both the final output and the correct answer were 3. After each sentence in the reasoning process, we appended `</think>The answer is:` to generate a response, examining the probabilities with which the model produced 2 or 3. Here, step 0 refers to the state before any reasoning has occurred, approximating the distribution of direct answers (note that the true direct answers were derived from different templates, whereas here only a single template is used); subsequent steps i (i > 0) correspond to each sentence in the model's normal reasoning procedure.

From the results in Figure 19, we can observe that, on average (Figure 19(a)), the model requires only one reasoning step to fully spell out the target word, and after which the influence of bias is no longer discernible in this measurement. (However, as demonstrated from multiple perspectives in the main text, bias still plays an underlying role in such cases.)

We further extract a "high-bias" subset from the these data and repeat the experiment (Figure 19(b)). In this subset, we observe a gradual decrease in the probability of decoding the biased answer and a corresponding increase in the probability of generating the correct answer throughout the reasoning process. This trend directly illustrates how the influence of bias diminishes progressively as the model performs more reasoning steps.

## L    STANDARD DEVIATIONS

Tables 9 and 10 present the standard deviations corresponding to the results reported in § 4.2 and § 5.1, respectively. We observe that the standard deviations are generally large, which can be attributed to significant variations in model output length. These variations are influenced by multiple factors, including problem complexity, internal bias, and parroting behavior. However, the standard deviations after applying the removing question intervention are consistently lower than those of the original outputs, demonstrating the improved stability with less internal bias affect.

Table 11 presents the standard deviations of the results of bias injection intervention in § 5.2. Table 12 presents the standard deviations of the results of mitigation trials in § 7.

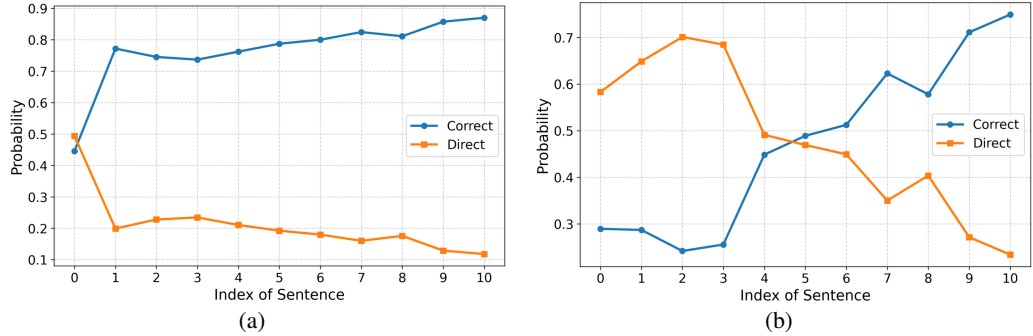

(a)                                                                                     (b)

Figure 19: The probabilities of decoding the biased answer and the correct answer after each sentence in the reasoning process: (a) average case; (b) high bias case.

Table 9: Standard deviations of results in § 4.2. For each model-dataset pair, the upper row shows the standard deviation for the high-deviation group, while the lower row corresponds to the low-deviation group. The relatively large standard deviations in the AIME datasets are due to its limited size.

| Model | CharCount(zh) | CharCount(en) | KnowLogic | AIME2024 | AIME2025 |
|---|---|---|---|---|---|
| DeepSeek-R1 | 557.6 | 267.3 | 3393.4 | 5850.4 | 7081.8 |
|  | 382.9 | 225.9 | 2800.1 | 6533.0 | 8030.2 |
| QwQ-32B | 778.8 | 566.5 | 2927.0 | 5886.5 | 4770.2 |
|  | 645.8 | 419.4 | 2847.9 | 5807.0 | 5681.1 |
| R1-Distill-Qwen-14B | 1173.8 | 440.2 | 3532.4 | 7211.2 | 5241.8 |
|  | 977.6 | 337.0 | 3151.1 | 6237.5 | 7789.7 |

Table 10: Standard deviations of the removing question intervention.

| Dataset | $L_{ori/rem}$ | $P_{first}$ |
|---|---|---|
| CharCount (en) | 422.9 / 248.2 | 134.2 |
| CharCount (zh) | 1086.0 / 589.8 | 186.1 |
| KnowLogic | 4224.8 / 4258.6 | 4311.7 |
| AIME 2024 | 6904.8 / 5630.9 | 3830.1 |
| AIME 2025 | 6762.8 / 6541.1 | 3037.6 |

Table 11: Standard deviations of the bias injection results.

| Setting | Length |
|---|---|
| *Random2Wrong* | 146.6 |
| *Low2Wrong* | 478.0 |
| *Random2Correct* | 684.3 |
| *High2Correct* | 561.9 |

Table 12: Standard deviations of the mitigation trials results.

| **CharCount** | $L_{low}$ | $L_{high}$ |
|---|---|---|
| R1-Distill-Qwen-14B | 977.6 | 1173.8 |
| + Remove | 684.3 | 554.2 |
| + FCS$_{DPO}$ | 520.3 | 625.2 |
| + FCS$_{SFT}$ | 457.0 | 549.5 |
| + SEAL | 421.9 | 780.2 |
| + PROBE | 673.6 | 886.5 |
| **AIME2024** | $L_{low}$ | $L_{high}$ |
| R1-Distill-Qwen-14B | 5850.4 | 6533.0 |
| + FCS$_{SFT}$ | 4666.4 | 6468.2 |

# M    LLM USAGE

The usage of LLMs in the writing of this paper was limited to improving linguistic clarity. Specifically, GPT-5[6] and DeepSeek (DeepSeek-AI et al., 2025a) were employed.

---

[6]https://openai.com/

