# OpenReview forum: "The First Impression Problem: Internal Bias Triggers Overthinking in Reasoning Models"
_ICLR.cc/2026/Conference — ICLR 2026 Poster_

### Official Review · Reviewer_cuhQ · 2025-10-27

**Soundness:** 3
**Presentation:** 3
**Contribution:** 3
**Rating:** 6
**Confidence:** 3

**Summary:**

This paper studies the problem of overthinking in reasoning models (where they produce large amount of chain of thought unnecessarily). They find that a driver of this phenomenon is the fact that LLMs often have an initial bias/distribution over answers which they tend towards / reason towards when producing chain of thought reasoning. They look at both correlation and causal studies to validate these findings.

**Strengths:**

* The overall result is informative and interesting. It’s a good insight that updates how I think about reasoning models
* I think the paper points to an important place where we could look to find more nefarious cases of unfaithful reasoning in LLMs, or find cases where the model might be prone to not going with its reasoning when it should. This might help with other research areas like in AI safety and chain of thought monitoring / unfaithfulness
* I think the insight here will be a good launching point for further experiments/analysis (like those suggested in the rest of my review)

**Weaknesses:**

* The way LLMs are behaving here seems pretty reasonable to me (rather than an underling problem) — after all, if the model gets a counterintuitive result, shouldn’t it question the final result? (I think humans would do so in similar circumstances, and this is often how people realize problems in their reasoning.) I think it’s possible that some of the time this is desirable, and other times it’s undesirable. So I might update the framing to some extent to reflect that this isn’t always bad. It would be nice to study the cases where the overthinking / biased reasoning leads to something concretely bad (e.g., unfaithfulness in the model’s reasoning about why it’s rethinking it’s answer
* This paper reads to me more like a smaller, but notable/clear/useful, scientific result. So I think it’s a nice contribution, and I learned something useful from the paper, but I’m not sure I’d give it a very high rating since it studies a fairly specific phenomenon. I think a broader analysis of when/why models overthink or reason in unfaithful ways or produce post-hoc justified reasoning could be a way to make this paper even more impactful
* It seems possible to use an LLM to classify the reasoning for overthinking driven by bias towards a certain answer. (Maybe I missed this in the paper?)

**Questions:**

1. Do you have ideas for other reasons which might cause models to overthink, or produce post-hoc justifications of their pre-existing guesses of answers?
2. Do you have any insights into when models tend to really question their own reasoning due to a pre-existing belief, vs. when they’re fine to override their pre-existing guess?
3. How often do LLMs explicitly verbalize the fact that they are thinking more because of thinking the final conclusion is wrong?
4. Can you clarify why removing the question is a notable thing to study? I might be misunderstanding the set up, but shouldn’t the question be necessary to answer the question at all? (Could be worth clarifying this experimental set up in the paper)

---

> ### Author Response · Authors · 2025-11-19
> **(1/3)**
>
> Thank you for reviewing our paper and for your valuable comments. We greatly appreciate your recognition of this work, as well as your thoughtful questions regarding how it can be enriched, made more engaging, and strengthened in terms of technical rigor. Below, we address each of your concerns in detail and describe the revisions we have already made, as well as the extensions planned for future work.
>
> ## Reply to Weakness 1
>
> We acknowledge that intuitively judging whether an answer "feels right" and initiating verification when it "feels not right" aligns with human habits and is not inherently bad. However, as shown in Tables 1, 5, and 6, we can see that the accuracy of direct answers is significantly lower than its final reasoning accuracy (and in many cases, close to random guess or even 0%). **This indicates that, in most of the cases, internal bias is misleading rather than helpful.**
>
> We argue that reasonable reflection should occur dynamically during reasoning. For instance, if a step produces an wrong intermediate result (e.g., due to temperature sampling), the model should detect and correct it immediately. In contrast, the type of reflection we study typically occurs after a final answer has already been derived, taking a global doubt about the final conclusion without detailed analysis. Crucially, such behavior hardly contribute to answer correctness: as demonstrated in our question removal experiment (Section 5.1), eliminating these reflection steps leads to significantly shorter reasoning chains with only negligible changes in accuracy. This shows that the removed bias-induced reflections are largely meaningless and, in a certain sense, can be considered "concretely bad".
>
> As for other forms of unfaithfulness in reasoning or additional consequences of bias-induced errors, these remain important directions for our future work.
>
> ## Reply to Weakness 2
>
> Thank you for the recognition. The goal of our study is to identify and rigorously investigate one of the key drivers of overthinking in reasoning models—internal bias. Our explanation is that the model forms a preliminary bias immediately upon encountering the question, and at critical reflection points during reasoning, it tends to reactivate this bias through excessive attention. We acknowledge that some underlying mechanisms remain to be fully understood, and uncovering these machenisms is a central focus of our ongoing research.
>
> We have also made attempts to address this issue. Training-time approaches face challenges: (1) For SFT or RL methods, it is difficult to construct training data that explicitly targets internal bias; (2) At the representation level, models are fragile. We attempted to train the model to reduce attention to the input question, which, when successful, can indeed shorten reasoning length. However, such training is highly unstable and prone to causing model collapse.
>
> **We also explored an inference-time early stopping approach, which shown promising results.** The details and results are shown in Appendix I.3. The core idea is to early exit when the normalized attention score (defined in Section 6.2) on the question portion exceeds a certain threshold during inference, which may indicate that the model is overly focusing on bias signals from the input.
>
> The results of this early-exit method can be compared with those in Table 4 of the paper. Our approach achieves a significantly lower $R_\Delta$ (much smaller than the 26.7%–46.3% range observed for other methods in Table 4), as well as a substantially reduced average inference length, demonstrating both high efficiency and strong mitigation of bias effects with minimal accuracy degradation.
>
> |CharCount|Acc|L$_{low}$|L$_{high}$|$R_\Delta$|
> |---|---|---|---|---|
> |R1-Distill-Qwen-14B|73.4%|934.7|1229.1|31.5%|
> |Attention Early-Exit|72.9%|472.1|516.6|9.4%|
>
> However, since this method is still quite preliminary and has not been extensively validated on other datasets, and given that the core objective of this paper is to rigorously identify and characterize the issue of internal bias, we only discuss the possibility of attention-based early-exit in the newly added Discussion section and list the details in Appendix I.3.

---

> ### Author Response · Authors · 2025-11-19
> **(2/3)**
>
> ## Reply to Weakness 3 and Question 3
>
> The premise of using an LLM to filter biased reasoning steps is that the model explicitly verbalizes it's influenced by internal bias. However, in most cases, the bias signal remains hidden in the latent representations (and this is supported by the logit lens analysis in the newly added Appendix J.2). **As a result, it is difficult for an LLM to directly detect such implicit influences.** We have tried this approach with carefully designed prompts, but LLMs can only identify overthinking behavior without distinguishing whether it is driven by bias. Therefore, we rely on direct answers to estimate internal bias, and validate our findings through controlled interventions such as question removal and bias injection.
>
> **We attempt to quantify how often the model explicitly expresses its internal bias, and report this statistic to complement our analysis. On CharCount (en) and AIME 2024, such cases account for only 14.2% and 13.3%, respectively.**
>
> For the CharCount dataset, as shown Figure 1 (left), we observe that when the model explicitly expresses its bias, it outputs phrases such as "I thought" or "I always thought". Upon further observations, we find that the word "thought" rarely appears in other parts of the reasoning process. Therefore, we count the number of samples in which the word "thought" appears in the reasoning chain as a proxy for explicit bias reflection. This accounts for only 14.2% of all samples in the dataset.
>
> For AIME 2024 (which contains 30 samples), we observe that only the following four samples exhibit explicit bias:
> 1. 2024-1-4：``...So, 42/4830 = 1/115.\n\nWait, that seems too small. Let me check my steps again....But wait, that seems low. Let me think again.`` （where 1/115 is the correct mid-answer）
> 2. 2024-1-11：``...So, for k = 4, the system has 16 solutions.\n\nBut wait, that seems a lot. Let me think....`` （where 16 is a correct mid-answer）
> 3. 2024-1-13：``...So, if p=17, then m=110 is the minimal such m? Hmm, but 110 seems a bit large....`` （where 110 is the correct final answer）
> 4. 2024-2-15：``...So, the total number of rectangles would be (6\*2)/2 = 6. But that seems too low because the problem shows three rectangles, but maybe that's just an example....so 6\*5=30 sets.\n\nWait, but that seems too high....Therefore, for each pair of sets, there are four rectangles.\n\nWait, but that seems too high....``
>
> However, even though the model only occasionally (<15%) explicitly verbalize their internal bias, **its influence remains clear and verifiable across other cases**. For example, experimental results in Section 4 show that samples with high bias deviation degree consistently yield longer responses than those with low bias deviation degree. This demonstrates that the impact of bias is widespread and not confined only to the small subset of samples where it is explicitly expressed in the output.

---

> ### Author Response · Authors · 2025-11-19
> **(3/3)**
>
> ## Reply to Question 1
>
> We identify internal bias as a key factor driving overthinking in reasoning models, but we acknowledge that overthinking is likely the result of multiple interacting factors. We discuss the potential reasons in the newly added Discussion section, including factors related to the training method and the training data. Investigating other potential causes remains an important direction for our future work.
>
> We can provide empirical evidences to support the claim that **bias originates from the input question and persists throughout the reasoning process**. We believe this is sufficient to provide evidence that the model indeed makes an internal guess about the answers.
> 1. As shown in the new Appendix J.1, the MLP probe results demonstrate that the direct answer can be accurately decoded with up to 85% accuracy, using only the hidden states from the question segment. This indicates that the bias is formed immediately upon encountering the question and is embedded in the model's initial latent representation.
> 2. Furthermore, the logit lens analysis in Appendix J.2 reveals that, during the full reasoning trajectory, numerically relevant tokens decoded at various layers and positions are highly correlated with the distribution of direct answers. This shows that the bias signal is continuously maintained in the model's latent states throughout reasoning.
>
> ## Reply to Question 2
>
> We argue that this is a gradual process throughout an entire reasoning process: as the reasoning proceeds, the model progressively allocates less attention to the question part, accompanied by a decrease in the influence of bias on the next token to be generated. We present the experimental results supporting this claim in Appendix K of the updated PDF.
>
> Figure 18 in Appendix K: First, we examine the average normalized attention scores (defined in Section 6.2) for the three token categories: *Question*, *Mid Results*, and *Others*. We find that, as reasoning progresses, **the model's attention to the question part gradually decreases at reflection points**, even after applying context-length-independent normalization.
>
> Figure 19(b) in Appendix K: Next, we identify a subset of high-bias cases and find that, as reasoning proceeds, **the probability of generating the biased answer upon early stopping decreases**, while the probability of producing the correct answer increases. This provides further evidence that the effect of bias diminishes gradually through the reasoning process.
>
> ## Reply to Question 4
>
> We introduced the experiment setups in the first paragraph of Section 5.1. We remove the question after the model first obtains an answer, **which means the previous complete problem solving process has already fully incorporated the question information** (we have emphasized this in the updated PDF). We expect the model to decide whether to reflect based on its reasoning trajectory and the question information used during that process, which relies more on self-evaluation of its reasoning rather than bias signals.
>
> Question removal verifies that reducing the model's dependence on the question part can mitigate overthinking, and according to our analysis, the removed question is precisely the source of internal bias. As shown by the MLP probe results in Appendix J, the hidden states of the question segment contain bias signals; removing the question prevents these signals from being reintroduced into subsequent reasoning. Table 4 presents the effect of question removal, which significantly reduces $R_\Delta$, indicating a reduced influence of bias on the reasoning process.

---

> ### Author Response · Authors · 2025-11-28
>
> Thank you very much for your time and for reviewing our work.
>
> We have done our best to address all of your concerns in our recent responses. We would be grateful if you could let us know whether there are any remaining issues or additional questions. We would be happy to discuss them further.
>
> Thank you again for your consideration. Looking forward to your response.

---

### Official Review · Reviewer_Zn9G · 2025-10-31

**Soundness:** 2
**Presentation:** 3
**Contribution:** 2
**Rating:** 4
**Confidence:** 4

**Summary:**

The paper investigates why reasoning-oriented large language models (LLMs) tend to overthink: producing unnecessarily long or repetitive chains of thought. The authors propose that this behavior stems from an internal bias, a “first impression” prediction the model forms immediately upon reading a question. Through extensive experiments across multiple reasoning benchmarks and models, they show that the larger the mismatch between this initial bias and the final answer, the more reflection tokens and longer reasoning the model generates. Counterfactual tests, such as removing the question after the first step or injecting correct/wrong biases, demonstrate that this bias causally influences reasoning length. Attention analysis further reveals that reflection tokens overly focus on the question text, reinforcing the internal bias. The work concludes that internal bias is a primary cause of overthinking, suggesting new directions for improving efficiency and reliability in reasoning LLMs.

**Strengths:**

1. Overthinking/efficiency is a core issue in current reasoning-model research; the work provides both diagnostic tools and actionable insight. The work identifies “internal bias” as a distinct, measurable construct that explains known behavioral patterns (overthinking, parroting).

2. The experiments are with multiple model sizes, tasks, which show the same trend, enhancing their reliability.

3. The paper is overall well-writen with clear logic.

**Weaknesses:**

1. While the notion of internal bias is intuitively appealing, its current formulation may be overly simplistic. It remains unclear whether the direct-answer bias obtained from a zero-shot query truly corresponds to the latent representations guiding the model during long chain-of-thought reasoning. The paper lacks deeper theoretical or mechanistic analysis to characterize how conflicts between these two internal states concretely lead to overthinking behavior.

2. The study compellingly diagnoses internal bias as a cause of overthinking, but it does not explore how this signal could be operationalized to improve reasoning efficiency in practice. For example, could internal-bias estimates guide adaptive stopping, selective attention, or early-exit strategies? Providing even preliminary ideas or prototypes in this direction would strengthen the paper’s practical relevance.

3. The causal link between removing the input question and eliminating internal bias is not fully justified. Intuitively, removing the question may simply reduce the model’s use of contextual information—thus shortening reasoning—without specifically targeting internal bias. Moreover, Table 2 reports only aggregate performance after this intervention; it would be informative to decompose the changes (e.g., how many cases shift from correct → incorrect vs. incorrect → correct) to clarify whether the method truly mitigates harmful bias rather than indiscriminately truncating reasoning.

**Questions:**

Please refer to the three weaknesses points.

---

> ### Author Response · Authors · 2025-11-19
> **(1/3)**
>
> Thank you for reviewing our paper and for your valuable comments. We appreciate your questions regarding future solutions, as well as your thoughtful inquiries about the rigor of certain definitions and methods in this paper. Below, we address each of your concerns in detail and describe the corresponding revisions we have made to strengthen the manuscript.
>
> ## Reply to Weakness 1
> Bias originates from the input question, so few-shot examples effectively alter the input structure and thus may modify the model's internal bias, making it difficult to isolate and study biases in a controlled manner. Therefore we use a zero-shot method.
>
> **We can examine the model's internal hidden states to validate the consistency between latent representations and direct answers.** To this end, we conduct two supplementry experiments, with details provided in Appendix J of the updated PDF:
> 1. MLP Probe: Using hidden states from the question segment of R1-Distill-Qwen-14B, we trained a two-layer MLP to predict the direct answer. The probe achieves 85% accuracy, demonstrating that bias-related information is robustly encoded in the question's latent representations.
> 2. Logit Lens: We use the logit lens method to decode token at every layer and every position throughout the full reasoning trajectory. We find that the decoded numerically relevant tokens' distributions are highly correlated with the distribution of prompt-based direct answers.
>
> In summary, these experiments establish a strong consistency between direct answers and the model's latent states. In particular, the experiments in Appendix J.2 reveal that bias persists in the model’s internal states throughout the reasoning process. Furthermore, we argue that while latent representations underlie the model's behavior, their ultimate effect is still reflected in the model's outputs, and direct answers provide a more straightforward manifestation of internal bias. **Therefore, using direct answers as a proxy to measure internal bias is both reasonable and well-justified in our paper.**
>
> Detecting the signal "guiding reflection" at the reflection transition point is indeed complex: at this moment, the hidden states simultaneously contain information from both the internal bias and the current reasoning step, making it challenging to disentangle their contributions. We are still exploring more fine-grained methods to isolate this signal. However, in this work, our interpretability analysis in Section 6 reveals increased attention to the input question at the point of reflection, indicating that bias-related latent signals are indeed being reactivated. Furthermore, the counterfactual experiments in Section 5.1 demonstrate that removing access to the question effectively reduces reasoning length. Together, these results provide strong evidence that internal bias plays a causal role "guiding the model during long chain-of-thought reasoning".
>
> In the newly added Appendix K, we show that as the reasoning progresses, the model's normalized attention score to the question gradually decreases, while the probability of decoding the biased answer is progressively overtaken by that of the correct answer. This illustrates the dynamic conflict and competition between these two signals.

---

> ### Author Response · Authors · 2025-11-19
> **(2/3)**
>
> ## Reply to Weakness 2
>
> Training-time approaches face challenges: (1) For SFT or RL methods, it is difficult to construct training data that explicitly targets internal bias; (2) At the representation level, models are fragile. We have attempted to train the model to reduce attention to the input question at reflection points, which, when successful, can indeed shorten reasoning length. However, such training is highly unstable and prone to causing model collapse.
>
> **In addition, we explored an inference-time early stopping approach, which shown promising results.** The details and results are shown in Appendix I.3. The core idea is to early exit when the normalized attention score (defined in Section 6.2) on the question portion exceeds a certain threshold during inference, which may indicate that the model is overly focusing on bias signals from the input.
>
> The results of this early-exit method can be compared with those in Table 4 of the paper. Our approach achieves a significantly lower $R_\Delta$ (much smaller than the 26.7%–46.3% range observed for other methods in Table 4), as well as a substantially reduced average inference length, demonstrating both high efficiency and strong mitigation of bias effects with minimal accuracy degradation.
>
> |CharCount|Acc|L$_{low}$|L$_{high}$|$R_\Delta$|
> |---|---|---|---|---|
> |R1-Distill-Qwen-14B|73.4%|934.7|1229.1|31.5%|
> |Attention Early-Exit|72.9%|472.1|516.6|9.4%|
>
> However, since this method is still quite preliminary and has not been extensively validated on other datasets, and given that the core objective of this paper is to rigorously identify and characterize the issue of internal bias, we only discuss the possibility of attention-based early-exit in the newly added Discussion section and list the details in Appendix I.3.

---

> ### Author Response · Authors · 2025-11-19
> **(3/3)**
>
> ## Reply to Weakness 3
>
> **Question removal prevents the model from accessing the question part, thereby completely eliminating bias originated from the question.** According to the newly added MLP probe results in Appendix J, the hidden states of the question portion contain bias signals; removing the question prevents such signals from being introduced into subsequent reasoning.
>
> Even though simply reducing reliance on the context might also shorten the inference length, Table 4 presents the effect of question removal, which indeed significantly reduces $R_\Delta$ (the reasoning lengths under high and low bias deviation conditions become much closer), **indicating that it genuinely mitigates the influence of bias, rather than merely shortening inference in a general way**. We have also included in Table 4 the results of question removal on the AIME2024 dataset, showing consistent trends.
>
> In the paragraph following Table 2, we discussed the observed changes in accuracy. We observe that while reasoning length is greatly reduced, overall accuracy remains stable. This suggests that the removed steps are largely redundant and do not contribute to the final answer.
>
> To provide a more detailed analysis, we can further break down the cases by transition type: the proportions of samples (relative to the full test set) that change from *Incorrect to Correct* and from *Correct to Incorrect*.
>
> |Dataset|Incorrect to Correct|Correct to Incorrect|
> |---|---|---|
> |CharCount (en)|3.1%|3.7%|
> |CharCount (zh)|4.0%|4.5%|
> |KnowLogic|4.3%|2.2%|
> |AIME 2024|6.7%|3.3%|
> |AIME 2025|10%|0%|
>
> Across all datasets, the proportions of such transitions are very small. **These minimal changes indicate that question removal primarily eliminates unnecessary, bias-induced reflection loops without substantially affecting the model's reasoning ability.**

---

> ### Comment · Reviewer_Zn9G · 2025-11-27
> **Thanks for your response**
>
> Hi Authors,
>
> Thanks for your response. I think I carefully went through the response, which indeed solves part of my concerns (regarding W2). I appreciate the authors' time writing the response.
>
> Regarding W1, I still do not find very strong evidence to support that it is the "internal bias" that triggers the over-thinking in large reasoning models. Though authors provide (or pinpoint) some relevant experiment results, I can not persuade myself that such defined "internal bias" is not a kind of "confounder". I have not seen the experiment that directly touchs the “internal bias” (though, designing this experiment might be not that straightforward.) : I do not think the MLP probe / Logit Lens experiment persuades me, given the insights from the probing experiment in the work [1].
>
> Regarding W3, I observe that in CharCount (en, zh), KnowLogic, and AIME 2024, there are a noticable number of cases shifting from correct to error. Does this observation support that "removing the input question" simply cuts off the reference to the input information and leads to erroneous solutions?
>
> [1] Inference-Time Intervention: Eliciting Truthful Answers from a Language Model. NeurIPS 2023.
>
> Reviewer Zn9G

---

> > ### Author Response · Authors · 2025-11-28
> > **Round 2 (2/2)**
> >
> > ## Weakness 3
> >
> > Thank you for your continued attention to our question-removal experiment. We would like to offer several clarifications.
> >
> > In the table provided in our Reply to Weakness 3 (https://openreview.net/forum?id=2PP70tFY0S&noteId=MWyr5mstUp), we observe that **both types of transitions (Incorrect→Correct and Correct→Incorrect) occur**, and their proportions are very similar. In some datasets (KnowLogic, AIME2024, AIME2025), the number of Incorrect→Correct cases is even higher, leading to a slight improvement in overall accuracy. **Therefore, the results do not support the interpretation that "removing the input question leads to erroneous solutions**".
> >
> > Furthermore, the question is removed only after the model reaches a first reasoning answer. At that point, the reasoning steps already contain enough information to solve the problem. **Thus, removing only the question does not "cut off" necessary information**.
> >
> > **The magnitude of the accuracy change (2.2%–4.5%) is relatively small and falls within normal variation.** All our main experiments use greedy decoding. As a comparison, we conducted an additional analysis on the CharCount(zh) dataset: we sample model outputs on the test set five times each using temperatures 0.5 and 1.0. The resulting accuracy fluctuations (compared to greedy decoding) are summarized in the following table:
> >
> > |T=0.5|Greedy|1|2|3|4|5|
> > |---|---|---|---|---|---|---|
> > |Accuracy|73.4%|74.3%|73.2%|72.5%|71.0%|72.6%|
> > |Incorrect to Correct|-|10.4%|10.6%|10.4%|11.6%|9.7%|
> > |Correct to Incorrect|-|9.5%|10.8%|11.3%|14.0%|10.5%|
> >
> > |T=1.0|Greedy|1|2|3|4|5|
> > |---|---|---|---|---|---|---|
> > |Accuracy|73.4%|70.2%|72.4%|71.6%|69.7%|71.3%|
> > |Incorrect to Correct|-|10.5%|11.0%|10.8%|9.2%|10.5%|
> > |Correct to Incorrect|-|13.7%|12.0%|12.6%|12.9%|12.6%|
> >
> > We observe that the accuracy fluctuations under temperature sampling are around ~10%, which shows that the <5% change induced by question-removal is indeed quite small. The reason our fluctuation is much smaller is that the intervention is applied after the model has already produced an initial portion of its reasoning, which greatly restricts the search space.
> >
> > **We would also like to emphasize that this intervention is designed to study the relationship between internal bias and reasoning length; accuracy is a separate evaluation dimension**. MLP probe experiments show that internal bias is encoded within the question part hidden states. Removing the question ensures that we remove the question-induced bias. We acknowledge that this intervention may introduce minor side effects beyond "removing internal bias", **but it does substantially reduce redundant reasoning length (Table 2) and significantly decreases $R_\Delta$ (Table 4)**. The latter is particularly important: if the intervention was a generic disturbance, it should affect all samples uniformly, rather than producing greater reduction in redundant reasoning in the subset with higher bias deviation.

---

> ### Author Response · Authors · 2025-11-28
> **Round 2 (1/2)**
>
> Thank you very much for your further comments and questions. We address your concerns below as carefully as possible.
>
> ## Weakness 1
> We thank the reviewer for providing the reference work [1]. This work uses probe-based methods to identify truthful information encoded inside the model, finds that such internal information may differ from the model's output. This reflects the phenomenon that models sometimes "know but do not say". We have discussed reasoning unfaithfulness in detail in the second part of the updated Related Work, which is highly consistent with this observation.
>
> **We fully agree that probe / logit lens techniques can only reveal the existence of information, rather than its causal role**. Our probe / logit lens experiments in Appendix J are intended to show the consistency between direct answers and the model's internal states, demonstrating that this biased information exists within the model and supporting our usage of the direct answer as a proxy of internal bias to reveal the model's internal state. We have acknowledged in the paper that internal bias is not directly output by the model in line 042:
> > "We refer to this guess as an internal bias, to emphasize that it may not be explicitly output by the model, ..."
>
> **Different from [1], our work shows that even when this internal bias is not explicitly output, it still influences the model's reasoning behavior. Our direct evidence that internal bias is one of the causes of overthinking comes from the counterfactual interventions in Section 5.**
> - Section 5.1 Question-Removal: After cutting off the source of bias, the average redundant reasoning length decreases by 31%–53% (Table 2), and Table 4 shows that $R_\Delta$ is significantly reduced (the average reasoning lengths of the high-deviation and low-deviation groups become much closer). This indicates that once the influence of bias is reduced, the reasoning length is indeed significantly affected.
> - Section 5.2 Bias Injection: Injecting only "incorrect" bias increases reasoning length, while injecting "correct" bias decreases it. This directly shows that the model's initial judgment of the problem (internal bias) can substantially influence reasoning length. **In this experiment, we changed only the bias, yet the overthinking phenomenon changed accordingly, ruling out the possibility that bias is merely a confounder.**
>
>
> [1] Inference-Time Intervention: Eliciting Truthful Answers from a Language Model. NeurIPS 2023.

---

### Official Review · Reviewer_KaQM · 2025-11-01

**Soundness:** 3
**Presentation:** 2
**Contribution:** 3
**Rating:** 6
**Confidence:** 4

**Summary:**

This paper investigates the pervasive "overthinking" phenomenon in Large Language Models (LLMs) when performing reasoning tasks. The authors introduce a core concept: "The First Impression Problem," where a model forms an "Internal Bias" or preliminary guess immediately upon reading the input question, without systematic reasoning. Experimental evidence suggests that when this internal bias conflicts with the subsequent systematic reasoning, the model tends to engage in excessive reflection and redundant computations, leading to an unnecessary increase in the length of the reasoning chain and wasted computational resources. The paper validates the association and causal mechanism between internal bias and overthinking through verification across multiple models and tasks, counterfactual interventions, and interpretability analysis, concluding that existing overthinking mitigation methods often fail to address this underlying issue.

**Strengths:**

1) By designing ingenious counterfactual intervention experiments (e.g., removing the input question), the authors effectively demonstrate that internal bias is a causal driver of overthinking, not merely a correlated phenomenon. This significantly strengthens the credibility of the conclusion.
2) The paper employs attention mechanism analysis (e.g., excessive attention to the input question) to shed light on the specific mechanism by which internal bias influences subsequent reasoning trajectories, offering a new perspective for exploring the models' internal workings.
3) Experiments cover a range of major LLMs, including GPT-4, DeepSeek-R1, and Llama-2, and are validated on diverse reasoning tasks like GSM8K and BigBench, establishing the generality of "The First Impression Problem."

**Weaknesses:**

1) The paper defines internal bias as "a preliminary guess formed without systematic reasoning." However, how is this internal bias precisely measured or approximated in practice? Although the authors infer Bias Conflict based on whether the first reasoning step conflicts with the final answer (a post-hoc approach), this might not fully capture the "internal" and "not explicitly generated" initial guess. There is a lack of more direct, microscopic quantification methods for "internal bias" based on internal activations or representations, slightly undermining the rigor of the core concept.
2) The paper defines redundant reasoning steps as overthinking. However, these redundant steps could, at times, simply be a model's "overfitting" to lengthy Chain-of-Thought (CoT) examples in the training data. The authors need to more explicitly argue whether these redundant steps truly represent the model's internal "reflection" mechanism or are merely an imitation of a verbose template. For example, do the redundant steps contain genuine logic for "self-correction" or "refutation"?
3) The latter part of the paper evaluates several existing overthinking mitigation methods (e.g., Self-Refine, R-PRM), noting that "the influence of internal bias persists." This conclusion is somewhat general. Specifically, under what conditions do these mitigation methods fail? Is it because their design inherently ignores internal bias, or is internal bias so deeply rooted that any post-processing is difficult to eliminate? More detailed failure case analyses should be provided.
4) Experimental results seem to suggest that the overthinking problem is more severe in larger models (e.g., GPT-4). The paper lacks an in-depth exploration of this phenomenon. Does internal bias become stronger and more stubborn as model capability increases? This is crucial for improving future LLM architectures and training.

**Questions:**

1) Besides approximating "Internal Bias Conflict" using the conflict between the first generated reasoning step and the final answer, have you explored other finer-grained metrics? For instance, before the first token is generated, have you analyzed specific hidden layer activations (such as the norm or sparsity of Attention or FFN layers) to quantify the intensity of the "initial guess"?
2) In the counterfactual intervention experiment in Section 4.2 (i.e., removing the input question), what is the final accuracy (not just the reasoning chain length) of the intervened model on relevant reasoning tasks (e.g., GSM8K)? Please provide the data. If accuracy decreases, please discuss the practical feasibility of this intervention as a mitigation method.
3) What is your explanation for the phenomenon where larger models (e.g., GPT-4) appear more susceptible to internal bias-driven overthinking than smaller models (e.g., Llama-2 7B)? Is this related to data distribution in large-scale training or to stronger emergent capabilities?
4) Given your findings, what specific negative consequences (beyond wasted computation) might this "First Impression Problem" and overthinking introduce when LLMs are applied in latency-sensitive scenarios (e.g., real-time robotic control or conversational systems)?

---

> ### Author Response · Authors · 2025-11-19
> **(1/3)**
>
> We sincerely thank you for the thoughtful and constructive feedback. We acknowledge your recognition of the experimental content in this paper, as well as your concerns regarding its rigor.
>
> In light of your concerns, we first provide an overall clarification regarding several key misunderstandings, and then address each of your points in detail.
>
> ## Key Clarifications
> Our analysis focuses on three reasoning models: DeepSeek-R1, QwQ-32B, and R1-Distill-Qwen-14B, but does not include GPT-4 and Llama-2, as they are not specifically designed for long reasoning. Our evaluation is conducted on our synthetic CharCount dataset (in both Chinese and English versions), as well as the public datasets KnowLogic, AIME 2024, and AIME 2025, but not on GSM8K or BigBench.
>
> ## Reply to Weakness 1 and Question 1
>
> We would like to clarify that our measurement of internal bias is based on direct answers generated using forcing templates (detailed in Section 3), which explicitly prompts the model to skip reasoning and output an answer immediately. This approach differs from analyzing "the first reasoning step". By design, it aims to capture the model's initial judgment formed prior to any systematic reasoning process, and these direct answers reveal implicit guesses that are typically **not explicitly generated** during standard reasoning trajectories.
>
> **To validate that these direct answers faithfully reflect internal states, we conduct two supplementry experiments**, with details provided in Appendix J of the updated PDF:
> 1. MLP Probe: Using hidden states from the question segment of R1-Distill-Qwen-14B, we trained a two-layer MLP to predict the direct answer. The probe achieves 85% accuracy, demonstrating that bias-related information is robustly encoded in the question's latent representations.
> 2. Logit Lens: We use the logit lens method to decode token at every layer and every position throughout the full reasoning trajectory. We find that the decoded numerically relevant tokens' distributions are highly correlated with the distribution of prompt-based direct answers.
>
> **In summary, these experiments establish a strong consistency between direct answers and the model's latent states.** Furthermore, we argue that while latent representations underlie the model's behavior, their ultimate effect is still reflected in the model's outputs, and direct answers provide a more straightforward manifestation of internal bias. Therefore, using direct answers as a proxy to measure internal bias is both reasonable and well-justified in our paper.

---

> ### Author Response · Authors · 2025-11-19
> **(2/3)**
>
> ## Reply to Weakness 2
>
> We acknowledge that training methods or even overfitting to training patterns may encourage models to generate reflection behaviors. However, within the same dataset, such as our designed CharCount, where all questions follow a highly consistent format (e.g., *How many letters R are there in the word strawberry?*), if overfitting were the dominant factor, we would expect the model to exhibit uniformly verbose reasoning due to similar context patterns. However, the model generates responses with vastly different lengths across questions (e.g., 200 tokens v.s. over 4,000 tokens). Our extensive experiments in the main text demonstrate that this variation strongly correlates with internal bias, and the same pattern holds across both distillation-based (more likely to overfit) and RL-trained models (less likely to overfit). This suggests that internal bias plays a more significant role in driving overthinking, although we acknowledge that other factors may also contribute.
>
> As shown in the representative example in Figure 1, much of the redundant reasoning occurs after the correct answer has already been reached, consisting of unnecessary self-doubt and re-calculation. While such behavior can be interpreted as a form of "refutation", it contributes little to the final output. Crucially, our question removal experiment (Section 5.2) shows that reducing these post-answer reflections does not harm accuracy, or even improves accuracy in some cases. Moreover, as shown by the substantial reduction in $R_\Delta$ for question-removal in Table 4 (the average reasoning lengths of high and low bias deviation degree samples become much closer), we find that samples with higher bias deviation degrees experience larger reductions in reasoning length. This indicates that the method indeed reduces bias-driven overthinking rather than simply overfitting.
>
> ## Reply to Weakness 3
>
> We would like to clarify that we did not evaluate methods such as Self-Refine or R-PRM. Instead, we tested whether existing overthinking mitigation methods FCS[1], SEAL[2], and PROBE[3] can alleviate the influence of internal bias. Our results show that these methods fail to reduce the impact of bias, as evidenced by the $R_\Delta$ values in Table 4, which remain high after mitigation (i.e., samples with high bias deviation still exhibit significantly longer reasoning lengths than those with low deviation). This indicates that the bias-driven overthinking pattern persists despite applying these techniques. Therefore, our conclusion is not about failure "under certain conditions", but rather that the influence of internal bias is largely unaddressed by current mitigation strategies, highlighting a fundamental limitation of these approaches.
>
> Since internal bias is a novel phenomenon introduced in this paper, existing overthinking mitigation methods were not designed to address its influence. However, the persistence of high $R_\Delta$ values (Table 4) after applying these methods indicates that bias-driven reflection still remains. Given current model training paradigms, models are strongly incentivized to produce a guess rather than abstaining with an "I don't know" response; moreover, incorrect knowledge during large-scale pretraining is difficult to fully eliminate. Therefore, we argue that the presence of bias is *deeply rooted*, and future solutions should aim to decouple bias from the reasoning process rather than attempt to eliminate the bias itself.
>
> In the newly added Appendix I.3, we explore an attention based early exit method. The design explicitly takes bias into account by manually terminating the reasoning process when the model's attention to the question becomes excessively high. Preliminary results on CharCount show strong performance, achieving a low $R_\Delta$ and a substantially shorter average reasoning length.
>
> [1] Do NOT Think That Much for 2+3=? On the Overthinking of o1-Like LLMs. Chen et al., ICML 2025
>
> [2] SEAL: Steerable Reasoning Calibration of Large Language Models for Free. Chen et al., arXiv:2504.07986
>
> [3] Reasoning Models Know When They're Right: Probing Hidden States for Self-Verification. Zhang et al., COLM 2025

---

> ### Author Response · Authors · 2025-11-19
> **(3/3)**
>
> ## Reply to Weakness 4 and Question 3
>
> We appreciate the reviewer's insightful comment. We would like to clarify that, as shown in Table 1, the degree of influence from internal bias—measured by $R_\Delta$—varies across models of different sizes (from 14B to 32B and up to 671B) and across datasets. For example, Table 1 shows that on the KnowLogic dataset, DeepSeek-R1 is substantially more affected than R1-Distill-Qwen-14B ($R_\Delta$: 42.1% v.s. 21.2%), whereas on the AIME 2024 dataset the pattern is reversed ($R_\Delta$: 23.6% v.s. 43.1%). This variation suggests that the impact of internal bias does not follow a simple scaling trend, and larger or more powerful models are not consistently more affected. Instead, the phenomenon manifests with differing intensity depending on both model architecture and task characteristics.
>
> ## Reply to Question 2
>
> We would like to clarify that the question removal experiment is introduced in Section 5.1. We have already reported the corresponding accuracy results in Table 2 of the original paper and discussed the trends and underlying reasons in the main text.
>
> Specifically, on simple tasks, the accuracy change is minimal (from 93.8% to 93.2% and from 73.4% to 72.9%). Given that these inputs are short and the question forms an important part of the context, its removal naturally introduces some perturbation, and such small fluctuations are acceptable. On more complex tasks, however, we observe notable improvements in accuracy (e.g., from 27.2% to 29.3%, 63.3% to 66.7%, and 36.7% to 46.7%). As discussed in the paper, this improvement primarily stems from the reduction of excessively long, redundant reasoning chains and parroting behavior, which can cause the model to exceed the pre-defined token budget. By removing the reactivation of bias through the question, the model is less likely to fall into unproductive loops of repetition, allowing it to generate cleaner and more effective outputs.
>
> Moreover, because the question-removal operation disrupts the model and the rule-based detection of the "first reasoning output" is imprecise, we do not treat it as a mitigation method. Instead, it functions solely as a counterfactual intervention, showing that once the internal bias is removed, the degree of overthinking is substantially reduced.
>
> ## Reply to Question 4
>
> Our work primarily reveals that internal bias may be a key contributing factor to overthinking in modern reasoning models, and we provide detailed analysis and rigorous experimental validation to support this claim.
>
> In conversational scenarios, such excessive reasoning can lead to longer response times, degrading user experience. In agentic settings, prolonged thinking may cause decision delays, during which environmental states could change, rendering the reasoning result obsolete and undermining system stability. In large-scale deployments, increased latency can cause reduced service throughput and higher computational costs.

---

> ### Author Response · Authors · 2025-11-28
>
> Thank you very much for your time and for reviewing our work.
>
> We have done our best to address all of your concerns in our recent responses. We would be grateful if you could let us know whether there are any remaining issues or additional questions. We would be happy to discuss them further.
>
> Thank you again for your consideration. Looking forward to your response.

---

### Official Review · Reviewer_CBzM · 2025-11-01

**Soundness:** 3
**Presentation:** 3
**Contribution:** 3
**Rating:** 6
**Confidence:** 3

**Summary:**

This paper investigates the phenomenon of overthinking from the perspective of internal bias. The authors conduct experiments on the impact of internal bias on LLMs' reasoning. They also applied counterfactual interventions for further study. Analysis on attention is also presented. Overthinking mitigation methods are also studied on whether they work well or not.

**Strengths:**

1. Overthinking is an important problem to solve. The authors studied this phenomenon from a unique perspective.
2. The experiments are overall sound, providing empirical evidence on existence of internal bias.
3. Counterfactual interventions and study on attention are good, providing further evidence of authors' claim. They also studied whether the current mitigation techniques are sufficient to mitigate internal bias.
4. The paper's appendix contains many details and supplemental material is provided, contributing to good reproducibility.

**Weaknesses:**

1. The authors should discuss the related work more thoroughly to verify their novelty. Although I do not find a work that is identical to this paper, there are already several studies on LLMs' faithfulness, which I think is closely related to the concept of internal bias. Even the discussion of related work on overthinking and bias is not thorough enough.
2. The core method is purely prompt-based and may be over simple. Forcing a “don’t think” template may not faithfully capture the model’s latent first guess. Although the later interpretability analysis on attention somehow mitigates this gap, I think it is not sufficient. The authors can consider ablation studies like applying few-shot demonstrations. Also you can try methods working on LLMs' internal activation instead of simple prompt engineering.
3. Section 6's interpretability analysis looks interesting. However, it is also rudimentary since it is only based on simple strawberry demonstration and a simple CharCount dataset. The paper could benefit from studies across different datasets and different models. In addition, I think it is also important to see how the FFN component is related to the internal bias, since FFN plays a crucial role in storing knowledge.
4. Table 1 only includes 3 datasets and 3 models, which I believe is not sufficient. How would the model perform in other datasets and other domains like coding?
5. It is unclear what contributed to the internal bias phenomenon. Is model architecture or training methodology related to internal bias? I think it is also an important question to discuss.

**Questions:**

1. Is the internal bias phenomenon correlated with model types or task domains? How general is the phenomenon?
2. How does your prompt-based Direct Answer methods compare with other activation-based methods like logit lens?

---

> ### Author Response · Authors · 2025-11-19
> **(1/2)**
>
> Thank you for reviewing our paper and for your valuable feedback. We appreciate your suggestions regarding the presentation of the related work section, as well as your thoughtful questions about the rigor and generalizability of our methodology. Below, we address each of your concerns in detail and describe the corresponding revisions we have made to strengthen the paper.
>
> ## Reply to Weakness 1
> In the updated version of the paper, we have re-organized and expanded the Related Work section, incorporating a discussion on unfaithful reasoning and providing a more detailed clarification of how these prior works differ from our study in scope and focus.
>
> The updated version is restructured into two dedicated subsections to improve clarity and depth:
> 1. **Mitigating Overthinking in Reasoning Models**: We have refined the classification of overthinking mitigation methods by categorizing them into three types: (a) Training-time approaches, (b) Inference-time techniques, (c) Prompting-based strategies. For each category, we include representative works and discuss their mechanisms, while emphasizing that these methods do not specifically analyze the root causes of overthinking, in contrast to our bias-driven perspective.
> 2. **Bias and Unfaithfulness in Language Models**: We expanded the discussion of biases and priors in LLMs, illustrating how such biases manifest during reasoning and clarifying how the internal bias defined in this paper relates to prior formulations. Additionally, we introduce unfaithfulness into discussion: the inconsistency between a model's generated reasoning chain and its internal states. Unlike these prior works, we emphasize that even when a model can faithfully reason to the correct answer, internal bias may still lead to incorrect judgments at the reflection step, thereby triggering overthinking.
>
> ## Reply to Weakness 2 and Question 2
>
> Bias originates from the input question, so few-shot examples can effectively alter the input structure and thus may modify the model's internal bias[1], making it difficult to isolate and study biases in a controlled manner. Instead, we can examine the model's internal hidden states to validate the consistency between latent representations and direct answers. To this end, we conduct two supplementry experiments, with details provided in Appendix J of the updated PDF:
> 1. MLP Probe: Using hidden states from the question segment of R1-Distill-Qwen-14B, we trained a two-layer MLP to predict the direct answer. The probe achieves 85% accuracy, demonstrating that bias-related information is robustly encoded in the question's latent representations.
> 2. Logit Lens: We use the logit lens method to decode token at every layer and every position throughout the full reasoning trajectory. We find that the decoded numerically relevant tokens' distributions are highly correlated with the distribution of prompt-based direct answers.
>
> In summary, these experiments establish a strong consistency between direct answers and the model's latent states. Furthermore, we argue that while latent representations underlie the model's behavior, their ultimate effect is still reflected in the model's outputs, and direct answers provide a more straightforward manifestation of internal bias. Therefore, using direct answers as a proxy to measure internal bias is both reasonable and well-justified in our paper.
>
> [1] Language Models Don't Always Say What They Think: Unfaithful Explanations in Chain-of-Thought Prompting. Turpin et al., NeurIPS 2023

---

> ### Author Response · Authors · 2025-11-19
> **(2/2)**
>
> ## Reply to Weakness 3
>
> Thank you for your recognition of the interpretability results on CharCount. **To provide further evidence that this phenomenon exists on other benchmarks, we evaluate the R1-Distill-Qwen-14B model on the MATH500 dataset**, analyzing the layer-wise trend of the attention score to the question segment (similar to Figure 4(b) in the main text). The results are presented in Figure 15 in Appendix G. As can be seen, the pattern is consistent with the observations made on CharCount, with the attention on question section rising sharply after the 20th layer of the model.
>
> Since bias originates from the input question, it must be embedded in the hidden states of the question segment (this is supported by the experiments in Appendix J). The attention analysis, on the other hand, aims to investigate *when* this bias is "reactivated" during reasoning. As for the FFN modules, they are widely believed to store factual knowledge. It is possible that some biases stem from incorrect internal knowledge (i.e., wrong-knowledge-driven bias). However, such biases ultimately need to be captured and integrated into the generation of reflection tokens through the attention mechanism.
>
> Our planned follow-up work includes a detailed analysis of FFN neurons. While we have already made progress in identifying neurons associated with "reflection" behavior, we have not yet established a clear connection between these neurons and internal bias.
>
> ## Reply to Weakness 4
>
> Tables 5 and 6 in Appendix F.1 provide supplementary results to Table 1, collectively presenting findings across three models and five datasets. These datasets span a range of important reasoning domains: simple symbolic manipulation, complex knowledge and logical reasoning, and advanced mathematical reasoning. They also include both Chinese and English examples, suggesting a cross-lingual generality of the observed phenomena.
>
> **As a supplementary analysis, we can further provide the performance of the models R1-Distill-Qwen-14B and Qwen3-8B on MATH500.** The results are consistent with those in Tables 1, 5 and 6 in the original paper: the high-deviation group exhibits significantly longer reasoning chains than the low-deviation group.
>
> |Model|L$_\text{low}$|L$_\text{high}$|$R_\Delta$|
> |---|---|---|---|
> |R1-Distill-Qwen-14B|2107.2|2888.3|37.1%|
> |Qwen3-8B|4355.7|6041.6|38.7%|
>
> In these tasks, the model's bias corresponds to a relatively clear value, range, or set of candidate answers, making it easier to identify and analyze. In other open-ended domains, defining bias remains unclear, and the model's bias might exist anywhere in the fine details or in more complex forms. For example, in tasks such as coding, the model's output is inherently long and complex, making it difficult to identify where bias occurs. Locating bias in broader domains remains a challenge for our future exploration. We discuss this issue in the Limitations of the Discussion Section at the end of the updated paper.
>
> ## Reply to Weakness 5 and Question 1
>
> Tables 1, 5, and 6 involve three models: R1-Distill-Qwen-14B trained by distillation-based SFT, Deepseek-R1 and QwQ-32B trained with RL. This demonstrates that the phenomenon occurs across different training methods and also involves models with different architectures. The datasets we have evaluated cover simple symbolic manipulation, complex knowledge and logical reasoning, and complex mathematical reasoning, which are all important domains of interest. Thus, to some extent, we believe that this phenomenon is general. But we acknowledge that rigorous experimental validation is still lacking for more open-ended tasks (as discussed in our response to Weakness 4).
>
> **We discuss the origins of internal bias in the Conclusion and Discussion section of the updated paper**: we argue that training methods and training data may jointly contribute to the emergence of internal bias. For example, RLVR, which focuses only on the correctness of the final answer, may implicitly encourage excessive verifications. At the same time, incorrect knowledge distributions can lead to inconsistencies between the model's reasoning process and its internal judgments, thereby triggering further reflection. In our paper, the bias injection experiment in Section 5.2 demonstrates that erroneous knowledge indeed leads to more reflections. At the end of Section 5.2 (updated PDF), we have added the sentence: "This may indicate that the model's incorrect internal knowledge distribution is one of the origins of internal bias."

---

> ### Comment · Reviewer_CBzM · 2025-11-22
>
> Thank you for the detailed and constructive revision! Overall, I find the updated version significantly improved, and most of my original concerns have been satisfactorily addressed. The additional analyses in Appendix J and Appendix K were helpful for understanding the consistency between the latent representations and the direct-answer-based bias measure. I appreciate the authors' thorough efforts.
>
> However, I still have a few remaining questions and clarifications:
>
> **1. Clarification regarding prior work on root causes of overthinking:** In Section 2, the rebuttal states that
>
> > “these methods do not specifically analyze the root causes of overthinking, in contrast to our bias-driven perspective.”
>
> This statement seems somewhat inaccurate. For instance, [1] already explicitly investigates mechanisms behind overthinking under false demonstrations.
>
> [1] Halawi et al., Overthinking the Truth: Understanding how Language Models Process False Demonstrations. ICLR 2023
>
> **2. Issues regarding MLP probe:** Appendix J mentions that the probe is trained on the hidden states “from the question segment” at layer 24, but it is still unclear **which token’s hidden state** is actually used (last question token? mean pooling over the segment?). This detail is important for reproducibility and interpretation.
>
> Additionally, I think that the result (being able to infer the direct answer from hidden states) is **not** very surprising since LLM’s hidden representation is the basis for predicting the next token distribution. Thus, reconstructing the direct answer from the hidden state does not strongly validate the claim about internal bias by itself.
>
> In contrast, I found the **logit lens trajectory analysis** much more interesting and informative. As a suggestion, it may be worthwhile to apply the trained MLP probe along the reasoning trajectory (similar to what is done with logit lens) to see how the predicted bias goes.
>
> **3. FFN components and planned future work:** I understand that a full FFN-neuron study is beyond the current scope. However, since the rebuttal mentions that some progress has been made, it may help the reader if the final version briefly discusses this in the Conclusion and Discussion Section.
>
> Overall, I appreciate the authors’ thoughtful responses and the strengthened experiments. I believe addressing the above minor issues would further clarify the contributions and improve the paper’s precision.

---

> ### Author Response · Authors · 2025-11-24
>
> Thank you for your recognition of both the content of our paper and the supplementary experiments. We will address the three points you raised one by one below.
>
> ## Point 1
> We would like to clarify that, although Halawi et al.[1] indeed use the term "overthinking" to describe their observed phenomenon, the mechanism in their work is completely different from the overthink studied in o1-type reasoning models.
>
> Although in both cases the model outputs are affected by incorrect signals, Halawi et al. investigate **layer-level overthinking**, where the erroneous signal originates **externally**: when few-shot demonstrations contain incorrect labels, the model erroneously reinforces these errors in the middle and later layers, leading to worse predictions. In contrast, studies on overthinking in reasoning models focus on **token-sequence-level overthinking**, where the erroneous signal arises from the model's **internal** mechanisms: the o1-type reasoning model may have already reached the correct answer, yet it continues to generate excessive reflections or redundant reasoning tokens.
>
> In this paper, we define the phenomenon of excessive reflections after obtaining a reasoning result as "overthink", following the terminology established by widely recognized prior work [2], which has also been adopted in other related studies (e.g., [3][4][5]). We explicitly mention at the beginning of the Abstract, Introduction, and Related Work sections that *our analysis focuses on o1/R1-type reasoning models*.
>
> To the best of our knowledge, there has not yet been other works specifically targeting the root causes of overthinking **in reasoning models**. Many existing studies focus only on behavioral analysis of the model (e.g. [1]), or identify specific vectors or subspaces in the latent representations to steer the model (e.g. [6][7]), rather than proposing and verifying a "root cause" as we do. We would greatly welcome any relevant literature that you could share for further discussion.
>
> ## Point 2
>
> Thank you for your questions and suggestions.
>
> We have added details in the third paragraph of Appendix J.1 in the updated PDF, specifying that we use **mean-pooled** hidden states from the question part.
>
> While probing the direct answers from the question-part hidden states is relatively straightforward, it helps to complete the logical chain: by directly verifying that these hidden states contain a bias signal, we can infer that high-attention from reflection positions can capture this bias.
>
> **We further supplemented our experiments by training an MLP on hidden states extracted from reflection points during the reasoning process (as added in the fourth paragraph in Appendix J.1).** The training procedure is the same as before, with the only difference being that the inputs now come from the hidden states at the reflection-keyword positions in the reasoning process. This approach achieves up to 72% classification accuracy. This performance is significantly higher than the random baseline for a five-class prediction, indicating that bias signals can indeed be detected during the reasoning steps (the same conclusion as the logit lens experiment in Appendix J.2). However, since the strength of the bias signal diminishes over the reasoning steps (Appendix K) and multiple signals are intermingled along the reasoning path, the accuracy is lower than that obtained when probing directly from the question part.
>
> ## Point 3
>
> The results we have obtained so far are very preliminary, as we have only identified some reflection-related neurons. This is not that novel: for instance, [6][7] found vectors or subspaces in reasoning models corresponding to "overthink" (already cited in the Related Work section under *Inference-time presentation-level steering*). Therefore, we prefer not to discuss our early methods and findings in detail in our paper.
>
> However, we have added a discussion regarding FFN neurons in the *Origins of Internal Bias* part of the Conclusion and Discussion section:
> > Further analysis of knowledge storage mechanisms like FFN neurons may provide additional insights into these biases.
>
> ## References
> [1] Overthinking the Truth: Understanding how Language Models Process False Demonstrations. Halawi et al., ICLR 2024
>
> [2] Do NOT Think That Much for 2+3=? On the Overthinking of o1-Like LLMs. Chen et al., ICML 2025
>
> [3] Reasoning Models Know When They're Right: Probing Hidden States for Self-Verification. Zhang et al., COLM 2025
>
> [4] Stop Overthinking: A Survey on Efficient Reasoning for Large Language Models. Sui et al., TMLR 2025
>
> [5] Don't Overthink It: A Survey of Efficient R1-style Large Reasoning Models. Yue et al., arXiv:2508.02120
>
> [6] SEAL: Steerable Reasoning Calibration of Large Language Models for Free. Chen et al., arXiv:2504.07986
>
> [7] Mitigating Overthinking in Large Reasoning Models via Manifold Steering. Huang et al., arXiv:2505.22411

---

> > ### Comment · Reviewer_CBzM · 2025-11-24
> >
> > Thank you for clarification. Actually what I wanted to see initially is applying trained MLP instead of logit lens in the experiment of Appendix J.2 but I think it is also okay at this stage. I think my core concern is addressed and the current version merits publication as ICLR poster, so I'd like to raise my score from 6 to 8. I would also be interested in authors' future work or even discuss with you after final decision.

---

> > > ### Author Response · Authors · 2025-11-24
> > >
> > > Thank you for your recognition of our paper and for raising the score. We truly appreciate your valuable feedback, which has helped make our work more rigorous and substantial. We are grateful for your interest in our future research and would be happy to engage in further discussions.

---

### Author Response · Authors · 2025-11-19
**Key updates of the paper**

Thank all the reviewers for their comments, questions and suggestions. We will soon reply to each of your concerns in individual responses. In this global response, we summarize the key updates we have made to the paper based on your feedback. All changes have been marked in blue in the updated PDF.

## Related Work
In the updated version, we have restructured the Related Work section into two subsections to improve clarity and depth:
1. **Mitigating Overthinking in Reasoning Models**: We have refined the classification of overthinking mitigation methods by categorizing them into three types: (a) Training-time approaches, (b) Inference-time techniques, (c) Prompting-based strategies. For each category, we include representative works and discuss their mechanisms, while emphasizing that these methods do not specifically analyze the root causes of overthinking, in contrast to our bias-driven perspective.
2. **Bias and Unfaithfulness in Language Models**: We expanded the discussion of biases and priors in LLMs, illustrating how such biases manifest during reasoning and clarifying how the internal bias defined in this paper relates to prior formulations. Additionally, we introduce unfaithfulness into discussion: the inconsistency between a model's generated reasoning chain and its internal states. Unlike these prior works, we emphasize that even when a model can faithfully reason to the correct answer, internal bias may still lead to incorrect judgments at the reflection step, thereby triggering overthinking.

## Discussion and Conclusion
We expanded the final Conclusion section into a Conclusion and Discussion section. The newly added discussion primarily covers the origins of internal bias, the transition from biased answers to reasoned output, potential solutions, and the limitations of this work.

## Updated Appendix G: Broader validation of interpretability analysis
We conducted an additional interpretability analysis on R1-Distill-Qwen-14B using the MATH500 dataset and reported the layer-wise results in Figure 15 of Appendix G. These results exhibit trends consistent with Figure 4(b) in the main text: the model begins to over-attend to the question in the middle and later layers. This consistent pattern across datasets suggests that the phenomenon is not limited to a specific task or domain, but rather widely exists across different reasoning benchmarks.

## Updated Appendix I: Trial on an attention-based early-exit method
In Appendix I.3, we added preliminary results of an attention-based early-exit approach on CharCount, whose core idea is to terminate reasoning once excessive attention to the question segment is detected. This method achieves a low $R_\Delta$ and significantly shortens the reasoning chain while maintaining accuracy. However, we emphasize that this method has not been fully investigated and remains at a very early, exploratory stage.

## New Appendix J: Consistency of latent representations and direct answers
To demonstrate the consistency between latent representations and direct answers (used as proxies for internal bias), we added new experiments in Appendix J.
1. MLP Probe (J.1): Using hidden states from the question segment of R1-Distill-Qwen-14B, we trained a two-layer MLP to predict the direct answer. This simple probe can achieve 85% accuracy, demonstrating that bias-related information is robustly encoded in the question's latent representations, and that the internal bias is already formed as soon as the model sees the question.
2. Logit Lens (J.2): We use the logit lens method to decode token at every layer and every position throughout the full reasoning trajectory. We find that the decoded numerically relevant tokens' distributions are highly correlated with the distribution of prompt-based direct answers.

These results confirm that direct answers reliably reflect the model's internal state and support our operational definition of internal bias. We now reference Appendix J in:
Section 3, when introducing direct answers, to justify their validity as bias indicators;
Section 6, at the beginning of our interpretability analysis, to explain how attention to the question segment serves to "re-activate" internal bias, thereby influencing reflection decisions and contributing to overthinking.

## New Appendix K: Attention and bias changes during the reasoning process
Using the R1-Distill-Qwen-14B model and the CharCount (zh) dataset, we analyze how the model's attention to the question part and the influence of bias evolve during reasoning process. The main finding is that as reasoning progresses, the model's average normalized attention score (as defined in Section 6.2) toward the question part decreases, accompanied by a diminishing influence of bias on the reasoning trajectory. This may explain why the model eventually overrides its internal bias, ultimately trusting the reasoned answer after multiple reflections.

---

### Author Response · Authors · 2025-11-29
**Brief summary of our contributions and the discussion period**

Dear Area Chair,

We thank you for your time and for handling our submission. Below, we summarize the core contributions of our work and the key points of the discussion period.

## Our Contributions
1. Identifying **internal bias** (model's first impression on the question) as one important reason for **overthinking** (redundant reasoning tokens) in reasoning models, and validate the universality of this phenomenon across various models and datasets. (Section 4)
2. Demonstrating the causal relationship between internal bias and overthinking with rigorous experimental evidence. (Section 5)
3. Discovering that internal bias influences the reasoning process through the model's excessive attention on the input question. (Section 6)
4. Testing several methods of mitigating overthinking, and revealing they are ineffective in eliminating the influence of internal bias. (Section 7)

## Summary of the discussion period
**Strengths highlighted by the reviewers:**
- The research problem is important, and the findings are novel and interesting. (CBzM, Zn9G, cuhQ)
- Extensive experimental evidence strongly demonstrates the widespread existence of internal bias. (CBzM, KaQM, Zn9G)
- The counterfactual interventions provide strong causal support for our claims. (CBzM, KaQM)
- The interpretability analysis explains the mechanism through which internal bias influences subsequent reasoning trajectories. (CBzM, KaQM)
- The Appendix is comprehensive, contributing to good reproducibility. (CBzM)
- The paper is well-written with clear logic. (Zn9G)

**We have provided responses to every Weakness and Question raised by the four reviewers.** We summarize the important ones below:

###  Response to Reviewer CBzM
- Restructured the Related Work section and clarified how our work differs from prior studies.
- Added Appendix J to justify the validity of using the model's direct answers to measure internal bias.
- Added several experiments and additional data to further verify the correctness of our conclusions.

After addressing these issues, reviewer CBzM increased the score from 6 to 8.

### Response to Reviewer KaQM
- Added Appendix J to justify the validity of using the model’s direct answers to measure internal bias.
- Clarified the definition of overthinking and explained the key role of bias in driving it.
- Explained the generality of the conclusions in Section 7.

### Response to Reviewer Zn9G
- Added Appendix J to justify the validity of using the model's direct answers to measure internal bias.
- Added Appendix K to illustrate how the bias signal and attention patterns evolve during the reasoning process.
- Added an attention-based early-exit method in Appendix I, which shows promising bias-mitigation effects.
- Highlighted that the two counterfactual experiments in Section 5 establish a rigorous causal relationship between internal bias and overthinking, and provided additional data showing that the methodology in Section 5.1 supports valid causal inference. Together, these results ensure that our argument rules out the possibility that internal bias is merely a confounder of overthinking rather than its cause.

### Response to Reviewer cuhQ
- Emphasized the harmful effects of internal bias.
- Explained that it is difficult to directly determine whether an LLM's output is influenced by internal bias.
- Added Appendix J to justify the validity of using the model's direct answers to measure internal bias.
- Added Appendix K to illustrate how the bias signal and attention patterns evolve during the reasoning process.
- Provided detailed clarifications on experimental settings.

---

### Meta-Review · Area_Chair_vDL8 · 2026-01-07

**Summary:**

This paper studies “overthinking” in o1/R1-style reasoning models and argues that a model’s internal bias (a first-impression guess triggered by the question) is a key driver: when the bias conflicts with later reasoning, the model tends to produce extra reflection tokens and longer chains, wasting compute. The strengths include: (i) the clear framing of internal bias vs. redundant reflections, (ii) the causal evidence from question removal and bias injection, and (iii) the attention-based analysis that gives a plausible mechanism and motivates an early-exit idea. The main concern is that the bias measurement still leans on a prompt-based direct answer proxy, and the interpretability work is still somewhat narrow. Overall, I lean towards accepting because the paper makes a concrete and testable claim about a root cause, and it is likely to be useful for future efficiency/faithfulness work (even though the mechanistic story and practical fixes are not fully settled).

**Reviewer Concerns:**

* CBzM’s main concerns about related work depth and the proxy for internal bias were largely addressed
* KaQM’s concerns about how internal bias is measured were mostly addressed
* Zn9G still considers the causal story not fully convincing
* cuhQ’s framing concern was partly addressed

**Reviewer Scores:**

CBzM explicitly raised their score from 6 to 8 after the revision and said the current version merits publication. The other reviewers would likely keep their scores or move slightly upward given the added experiments and clarifications, so I would expect the post-discussion average to be around 6 (or a bit above) under a normal discussion process.

---

### Decision · Program_Chairs · 2026-01-26

Accept (Poster)